# A synthetic antibiotic class with a deeply-optimized design for overcoming bacterial resistance

Jin Feng[1], Youle Zheng[2], Wanqing Ma[2], Defeng Weng[2], Dapeng Peng[1,2], Yindi Xu[3], Zhifang Wang[3] & Xu Wang [1,2] ✉

The lack of new drugs that are effective against antibiotic-resistant bacteria has caused increasing concern in global public health. Based on this study, we report development of a modified antimicrobial drug through structure-based drug design (SBDD) and modular synthesis. The optimal modified compound, F8, was identified, which demonstrated in vitro and in vivo broad-spectrum antibacterial activity against drug-resistant bacteria and effectively mitigated the development of resistance. F8 exhibits significant bactericidal activity against bacteria resistant to antibiotics such as methicillin, polymyxin B, florfenicol (FLO), doxycycline, ampicillin and sulfamethoxazole. In a mouse model of drug-resistant bacteremia, F8 was found to increase survival and significantly reduce bacterial load in infected mice. Multi-omics analysis (transcriptomics, proteomics, and metabolomics) have indicated that ornithine carbamoyl transferase (arcB) is a antimicrobial target of F8. Further molecular docking, Isothermal Titration Calorimetry (ITC), and Differential Scanning Fluorimetry (DSF) studies verified arcB as a effective target for F8. Finally, mechanistic studies suggest that F8 competitively binds to arcB, disrupting the bacterial cell membrane and inducing a certain degree of oxidative damage. Here, we report F8 as a promising candidate drug for the development of antibiotic formulations to combat antibiotic-resistant bacteria-associated infections.

The rapid increase of drug-resistant pathogens, particularly the emergence of superbugs, poses a significant threat to public health[1–3]. The discovery and clinical application of antibiotics undoubtedly mark a milestone in human and modern medical history. They have driven advancements in various medical practices and are indispensable life-saving weapons[4–7]; however, the rise and spread of bacteria that have developed resistance to most or all existing antibiotics have raised concerns about an impending global infectious disease crisis[1,8–11]. In the United States alone, there are over 2.8 million antibiotic-resistant infections each year, resulting in more than 35,000 deaths[12]. In Europe, antibiotic resistance leads to about 33,000 deaths per year, with estimated hospital costs exceeding €900 million[12]. Any plan to address this issue relies on the discovery of new antibiotics that are effective against modern bacterial pathogens, making the development of antibiotics with unique mechanisms crucial[1,13,14]. The development of antibiotics is continuously challenging nowadays, leading to the rarity of new antibiotics being introduced clinically. Therefore, there is an urgent need

[1]National Reference Laboratory of Veterinary Drug Residues (HZAU) and MAO Key Laboratory for Detection of Veterinary Drug Residues, Huazhong Agricultural University, Wuhan, Hubei, China. [2]MAO Laboratory for Risk Assessment of Quality and Safety of Livestock and Poultry Products, Huazhong Agricultural University, Wuhan, Hubei, China. [3]Institute of Animal Husbandry and Veterinary Research, Henan Academy of Agricultural Sciences, Zhengzhou, Henan, China. ✉e-mail: wangxu@mail.hzau.edu.cn

to identify new targets and drugs to fill the gaps in antibiotic discovery and development, and to combat bacterial infections.

Structure-based drug design (SBDD) on the ribosome has encountered significant challenges, including low-resolution structural data and the lack of computational tools capable of effectively handling large RNA-based binding sites[15–17]. Early structural studies of the ribosome provided limited details, making it difficult to accurately model drug interactions and predict binding affinities. Before resolving the structures of ribosomal subunits and their complexes with known antibiotics, computational design of novel antibacterial agents focused primarily on ligands. These issues once hindered the progress of amphenicols' research. However, advancements in modern structural techniques and the development of new computational methods have now made structure-based drug design feasible, thereby greatly revitalizing this field of research. Furthermore, compared to combination therapy, SBDD better addresses issues of differential bioavailability, pharmacokinetics, and metabolism, resulting in improved therapeutic safety and avoidance of drug-drug interactions.

To find compounds with potent antimicrobial activity and a reduced propensity for resistance, this study employed the approach of SBDD and modular synthesis, leading to the discovery of the optimized modified compound, F8. We found that F8 exhibited in vitro and in vivo antibacterial activity, and through multi-omics analysis (transcriptomics, proteomics, and metabolomics), we inferred that the antibacterial target of F8 may be the key enzyme of arcB in the arginine degradation pathway. This study evaluated the therapeutic effects of F8 both in vitro and in vivo. F8 demonstrates significant antibacterial activity against both Gram-positive and Gram-negative bacteria, with a minimum inhibitory concentration (MIC) range of 2–8 μM. In a mouse model of FLO-resistant *S. aureus* bacteraemia, the bacterial burden in tissues was significantly reduced ($Log_{10}$ CFU/mL as low as 2–3), and the survival rate of the mice within 72 h is as high as 50% (all deaths within 24 h were in the control group). Furthermore, the antibacterial mechanism of F8 targeting arcB was systematically investigated. Here, we show that F8 competitively binds to arcB, thereby activating multiple pathways to exert its antimicrobial effects. The results of this study provides a foundation and strategy for the development of antimicrobial drugs, and reaffirmed the ability of chemical synthesis to supplement our antibiotic arsenal, providing broad-spectrum drugs that can overcome increasingly severe drug resistance mechanisms.

## Results

### Design and synthesis of broad-spectrum antibacterial small molecule compounds

In order to find compounds with effective antibacterial activity and that are less likely to develop resistance, we endeavoured to design and synthesize broad-spectrum antibacterial small-molecule compounds using computer assisted drug design (CADD)[18]. Guided by the SBDD and modular synthesis, we chose FLO, a compound with simple structure, low-molecular weight, and easy synthesis, as our main skeleton structure. With peptidyl transferase centre (PTC) of the bacterial 50 S ribosome subunit as the target, the key area of FLO, we applied CADD and developed a modular synthesis route. By forming an ester bond to connect these components, we produced a variety of antibacterial candidate drugs. Therefore, we studied a wide range of structural changes and evaluated their antibacterial activities.

First, we conducted molecular docking studies using the crystal structure of the PTC region of the 50 S subunit from the protein data bank (PDB: 6c4h)[19]. We then modelled the binding pocket of the PTC region of the 50 S subunit ("Methods") and established an atomic property field that reflects preferences for various atomic properties at each point in space (Fig. 1a). The structure-activity relationship of FLO has shown that its main scaffold occupies the conserved region of the binding pocket of the 50 S subunit's PTC region, therefore, we have chosen to keep this region static and instead structurally modified the

β-hydroxy position. The hydroxyl group provides a considerable advantage for the synthesis of small molecules. Even for the novel compounds generated, it must conform to a mechanism that can be easily synthesized, otherwise, it would have minimal significant broad influences[20].

Next, we selected various simple small molecule structures, including tricyclic, tetracyclic, pentacyclic, and hexacyclic structures, to connect with the skeleton structure (ChemDraw software), generating over a hundred compound structures for automated molecular docking. According to the scoring of SYBYL-X, some of the lead compounds exhibited improved binding power, reduced intermolecular collision force, and tighter binding with the receptor protein when compared to the skeleton structure (Fig. 1b and Supplementary Table 1). Based on docking simulations, ligands with smaller fragments have lower crash values, suggesting that structures with smaller spatial conformation within the binding pocket can lower the level of internal self-collisions that the ligand might experience, thereby enhancing the stability of binding. Crash value is the degree of improper docking between the ligand and the receptor protein. Crash scores close to 0 are favorable. Among these studies defined within relatively narrow parameters, we comprehensively selected 11 compounds with the best scores (Supplementary Table 1), total scores and polarity all higher than the total scores and polarity of the scaffold compounds (score = 5.1467), and with good binding specificity and affinity. These were then visually compared and synthesized against the PTC region of the 50 S subunit. The structure was confirmed by LCMS-IT-TOF, $^1$H-NMR, and $^{13}$C-NMR measurements (Fig. 1c, Supplementary Table 2, and Supplementary Notes).

We identified structure 8 (F8, 3-(1-Piperidinyl) propanoic acid) as the optimal modification, which further enhanced the antibacterial effects against pathogens such as *E. coli*, *S. aureus*, *S. typhi*, *P. multocida*, and *H. parasuis*, and imparted measurable activity against the particularly challenging *P. aeruginosa*. As shown in Table 1, structures 1, 2, 6, 7, 8, and 14 exhibit good antibacterial effects. When small molecule fragments consist of tetracyclic and pentacyclic structures, including thiazole and cyclobutane classes (structures 1, 2, and 7), their antibacterial activity is weaker compared to the skeleton structure. When the small molecule fragments feature hexacyclic structures (structure 14), the antibacterial activity is similar to the skeleton structure. Further using hexacyclic structures, we found that quinoline and morpholine structures (structures 11 and 13) virtually lose their antibacterial activities. We found that in general, short chain small fragments are more active than their homologues. Furthermore, it was found that the piperidine structure (structure 8) is sensitive to antibacterial activity and antibacterial spectra, and its effect is superior to that of the skeleton structure. Docking simulation shows that F8 can extend to a deeper position at the end of the PTC region of the 50 S subunit (Fig. 1b). We observed changes in the binding amino acid sites, with a greater number of hydrogen bonds between F8 and the target involved in ligand-protein binding than the skeleton structure, which may directly lead to an increase in binding affinity. Moreover, F8 binds near U2585 (key site), which may compete with the substrate of the a-site (one of the core functional regions of the ribosome), interrupting the binding of the a-site with tRNA and preventing the formation of peptide bonds, thereby playing a crucial role. The software also scored F8 (score = 6.1738) higher than the scaffold structure (score = 5.1467) (Supplementary Table 1), providing support for our design strategy.

### F8 is a candidate antibiotic

The chemical structure of F8 is shown in Fig. 1c. To evaluate its in vitro antibacterial activity, we determined the MIC against various bacteria, including both Gram-negative and Gram-positive bacteria. F8 exhibited significant antibacterial activity against both Gram-positive and Gram-negative bacteria, with MIC values ranging from 2 to 8 μM.

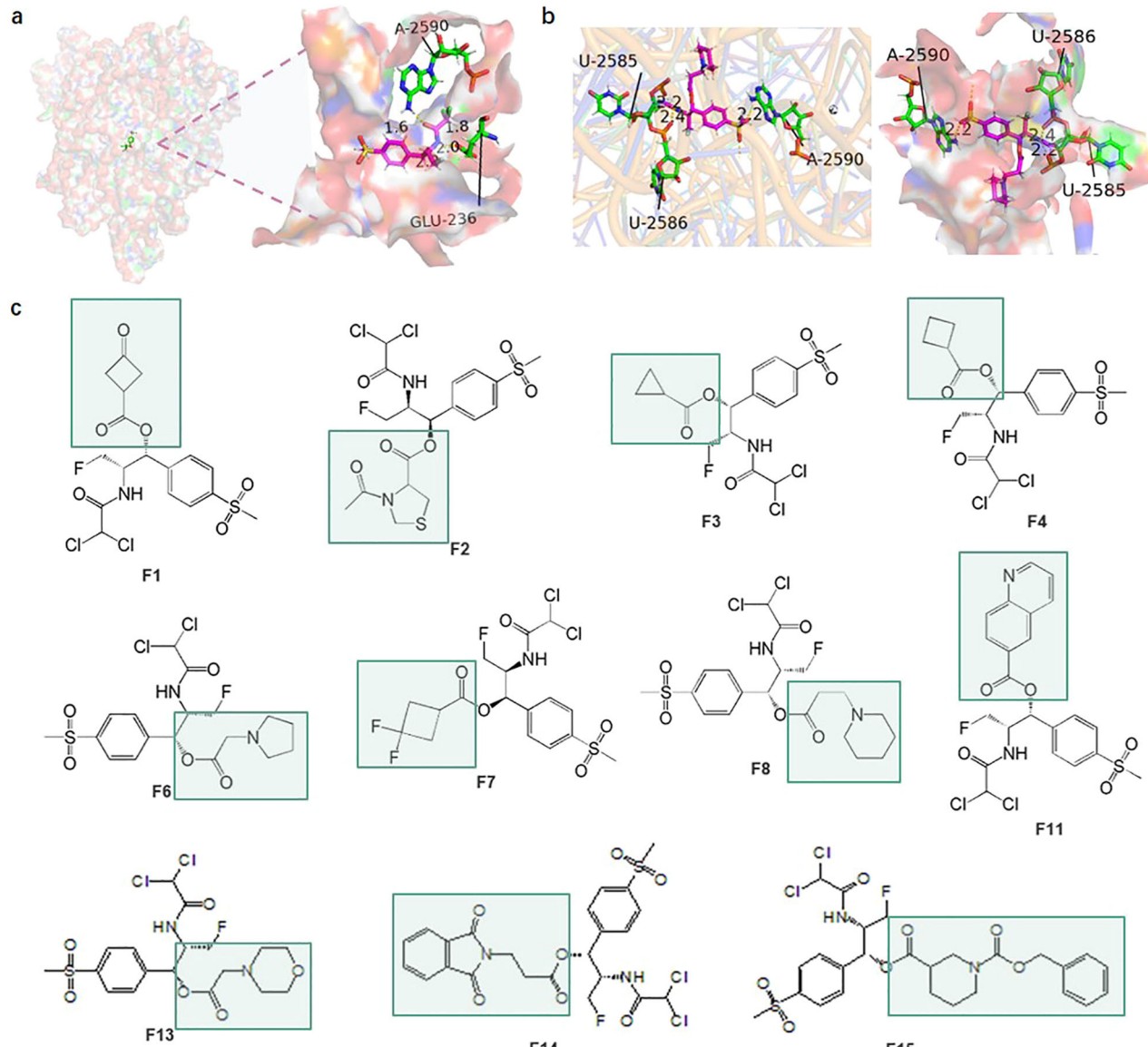

**Fig. 1 | Discovery of F8. a** Simulated image of the 4 Å drug-binding domain (DBD) of FLO in the peptidyl transferase centre (PTC) of the 50 s subunit generated by PyMOL software (PDB:6c4h), with residues GLU-236, A-2590, and C-2606 displayed in magenta. **b** Simulated image of the 4 Å DBD of F8 in the PTC of the 50 s subunit produced by PyMOL software, showing residues U-2585, U-2586, and A-2590 in green. **c** Structures of compounds.

Measurable activity was noted against the particularly challenging *P. aeruginosa*, with an MIC of 128 µM. F8 exhibits significant bactericidal activity against methicillin-resistant *S. aureus* (MRSA), polymyxin B-resistant *E. hormaechei*, FLO-resistant *S. suis*, FLO-resistant *H. parasuis*, doxycycline-resistant *S. typhi*, ampicillin-resistant *S. typhi* and sulfamethoxazole-resistant *S. typhi*, among other drug-resistant bacteria (Table 2). Additionally, we used a representative Gram-positive bacterium, *S. aureus*, and a Gram-negative bacterium, *E. coli*, as models to investigate the antibacterial activity and mechanism of action of F8.

To confirm its biological activity, the agar diffusion method was employed to test the antibacterial activity of F8 against *S. aureus* and *E. coli*. We observed that F8 consistently inhibited the growth of the pathogens, displaying distinct inhibition zones (Supplementary Fig. 1a). The growth inhibition effect of F8 on *S. aureus* and *E. coli* was further assessed (Supplementary Fig. 1b, c), and growth curves demonstrated that F8 effectively suppressed *S. aureus* and *E. coli* at 4×MIC, 2×MIC, and MIC, notably diminishing the growth of *S. aureus* at half the MIC concentration. Additionally, we monitored the bacterial

viability of *S. aureus* and *E. coli* exposed to different concentrations of F8 at various time points (Supplementary Fig. 1d–g), and found that F8 at 4×MIC, 2×MIC, and MIC could kill the bacteria within 8 h and eliminate the majority of bacteria for an extended period. Based on the minimum bactericidal concentration (MBC) (Table 3) and MIC values, we confirmed F8 as an effective candidate antibacterial agent.

To further evaluate the in vivo therapeutic potential of F8, we used a murine sepsis model induced by the intraperitoneal injection of *S. aureus* to test the therapeutic effect of F8 on bacterial infection. Bacterial colony counts revealed that F8 at 60 mg/kg could effectively reduce bacterial load in the liver, spleen, kidneys, and blood (Fig. 2a–d). The *S. aureus* infection group displayed extensive tubular necrosis with an indistinct structure in kidney tissues (cortical and medullary) (Fig. 2e). Inflammatory cell infiltration centred on granulocytes within the tubules, along with necrosis and conglomeration of inflammatory cells, significant tubular dilation, and focal haemorrhages were also notable in the infected group. Hepatic cells demonstrated vacuolar degeneration and inflammatory cell infiltration, with

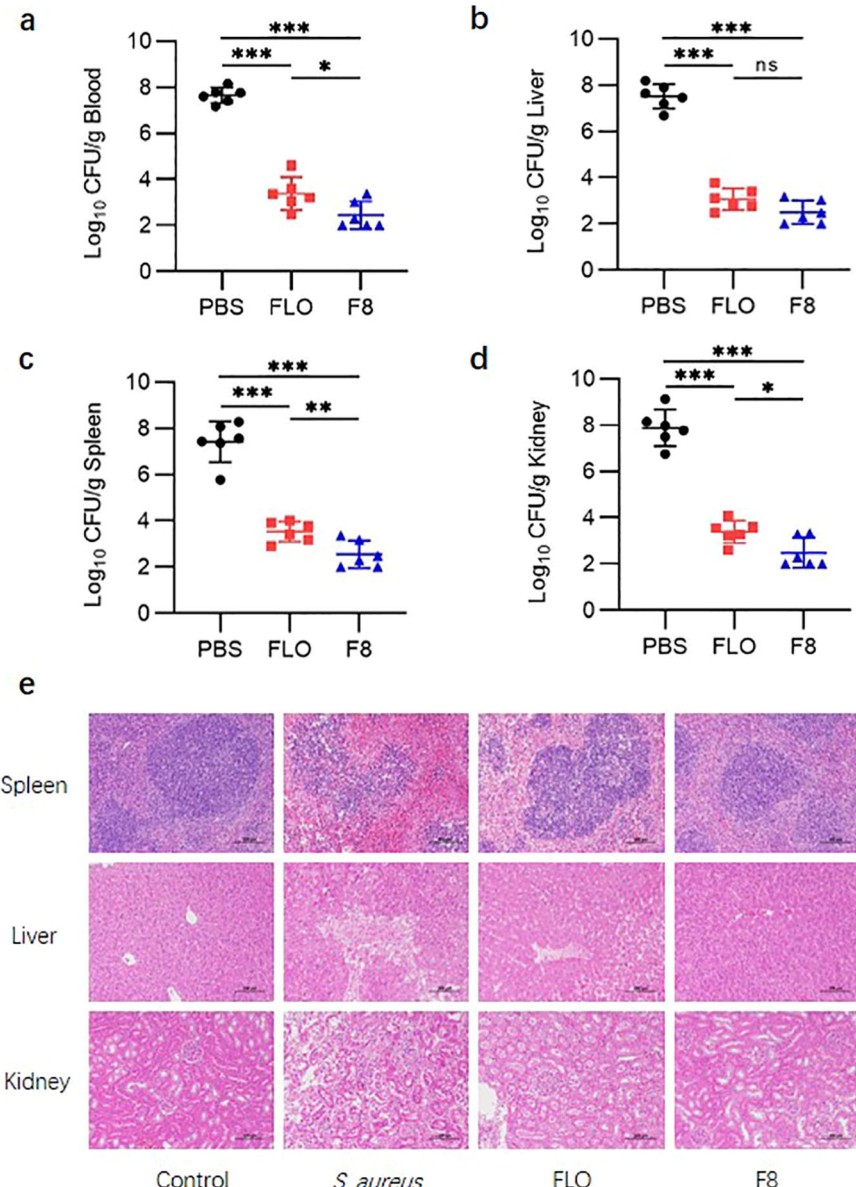

**Fig. 2 | Antimicrobial activity of F8 in vivo. a–d** Bacterial load in the blood, spleen, and kidney in the different treatment groups in a mouse *S. aureus* bacteraemia model (*n* = 6). Statistics: two tailed *t* test. Mean with SD. **a** (*P* < 0.0001 (FLO-PBS), *P* < 0.0001 (F8-PBS), *P* = 0.0332 (F8-FLO)). **b** (*P* < 0.0001 (FLO-PBS), *P* < 0.0001 (F8-PBS), *P* = 0.0727 (F8-FLO)). **c** (*P* < 0.0001 (FLO-PBS), *P* < 0.0001 (F8-PBS), *P* = 0.0084 (F8-FLO)). **d** (*P* < 0.0001 (FLO-PBS), *P* < 0.0001 (F8-PBS), *P* = 0.0217 (F8-FLO)). ns, not significant, \**P* < 0.05, \*\**P* < 0.01, \*\*\**P* < 0.001. **e** Histopathological assessment of the liver, spleen, and kidney in the different treatment groups in a mouse *S. aureus* bacteraemia model. Scale bar, 100 μm. Source data are provided as a Source Data file.

visible bleeding within vessels. The boundary between the red pulp and white pulp in the spleen was indistinct, with an increase in the number of neutrophils and macrophages within the red pulp. However, in the F8 treatment group, nearly no significant pathological changes were observed.

To further characterize the potential of F8 as an antimicrobial agent, we also investigated the effect of F8 on organ damage. Results showed that no notable abnormalities were observed in the liver, kidneys, and intestines in the F8-treated group at 1500 mg/kg, with no significant pathological changes in the spleen, kidneys, and bone marrow (Fig. 3a, b, i and Supplementary Fig. 2a, b); however, the positive control FLO group exhibited minor vacuolar degeneration and inflammatory cell infiltration in hepatic cells, with mild bleeding visible inside vessels. Kidney structure was abnormal with indistinct boundaries, a disappearance of tubular lumens, and glomerular atrophic degeneration. Extensive fibrosis with slight necrotic degeneration and shedding was observed in the intestinal tissues. Thymic medullary lymphocytes showed relatively loose arrangements with serum exudation, the number of lymphocytes was lower than in the control group, and minor inflammatory cell infiltration was observed within the medulla. The boundary between the red pulp and white pulp in the spleen was indistinct, with increased numbers of neutrophils and macrophages in the red pulp (Fig. 3i). ELISA results showed that there was no significant difference in the blood levels of IL-6 between the F8-treated group and the control group, but there was a significant difference in the levels of IL-2 and Hsp70 (*P* < 0.05) (Fig. 3c–e). As a highly conserved stress protein, Hsp70 significantly increases in expression level when exposed to harmful stimuli or stress factors, playing a crucial role in immune regulation. Compared with the control group, the concentration of Hsp70 in the thymus and spleen of the F8 group showed no significant difference, but significantly increased in the bone marrow (Fig. 3f–h). In addition, there

**Table 1 | MIC of broad-spectrum antibacterial small molecule compounds based on SBDD and modular synthesis**

| Organism | MIC (µM) | | | | | | | | | | | |
|---|---|---|---|---|---|---|---|---|---|---|---|---|
| | FLO | F1 | F2 | F3 | F4 | F6 | F7 | F8 | F11 | F13 | F14 | F15 |
| *E. coli* | 8 | 16 | 32 | >128 | >128 | 16 | 16 | 8 | >128 | >128 | 16 | >128 |
| *S. typhi* | 4 | 16 | 32 | >128 | >128 | 8 | 16 | 2 | >128 | >128 | 8 | >128 |
| *P. aeruginosa* | >128 | >128 | >128 | >128 | >128 | 128 | >128 | 128 | >128 | >128 | >128 | >128 |
| *S. aureus* | 16 | 32 | 64 | >128 | >128 | 16 | 32 | 8 | >128 | >128 | 16 | >128 |
| *B. subtilis* | 4 | 16 | 16 | >128 | 128 | 4 | 4 | 2 | >128 | >128 | 4 | >128 |
| *E. faecalis* | 16 | 16 | 32 | >128 | 128 | 8 | 8 | 4 | >128 | >128 | 8 | >128 |
| *P. multocida* | 8 | 16 | 8 | >128 | 64 | 4 | 4 | 2 | >128 | >128 | 2 | >128 |
| *A. pleuropneumoniae* | 8 | 32 | 32 | >128 | 128 | 4 | 16 | 4 | >128 | >128 | 8 | >128 |
| *S. suis* | 16 | 64 | 64 | >128 | >128 | 16 | 16 | 8 | >128 | >128 | 32 | >128 |
| *H. parasuis* | 32 | 64 | 64 | >128 | >128 | 16 | 32 | 8 | >128 | >128 | 16 | >128 |

*MIC* Minimum Inhibitory Concentration, *FLO* florfenicol.
The MICs of antibiotics were determined by the broth microdilution method (see "Methods") following the Clinical and Laboratory Standards Institute (CLSI) guidelines. The assay was done three times to confirm results (*n* = 3). Source data are provided as a Source Data file.

**Table 2 | MIC of F8 against antibiotic-resistant strains**

| Organism | MIC (µM) | |
|---|---|---|
| | F8 | FLO |
| Methicillin-resistant *S. aureus* B1-1 | 8 | 32 |
| Polymyxin B-resistant *E. hormaechei* wb 4 | 16 | 128 |
| Florfenicol-resistant *S. suis* 1136 | 8 | 64 |
| Florfenicol-resistant *S. suis* 1194 | 4 | 16 |
| Florfenicol-resistant *S. suis* 1197 | 8 | 32 |
| Florfenicol-resistant *S. suis* 1655 | 8 | 32 |
| Florfenicol-resistant *S. suis* 1658 | 4 | 32 |
| Florfenicol-resistant *S. suis* 1669 | 16 | 64 |
| Florfenicol-resistant *H. parasuis* 1565 | 8 | 32 |
| Florfenicol-resistant *H. parasuis* 1614 | 4 | 16 |
| Florfenicol-resistant *H. parasuis* 1651 | 8 | 32 |
| Ampicillin-resistant *S. typhi* BYG 9 | 4 | 64 |
| Sulfamethoxazole-resistant *S. typhi* BYG 21 | 4 | 32 |
| Doxycycline-resistant *S. typhi* BYG 11 | 4 | 32 |
| Doxycycline-resistant *S. typhi* BYG 25 | 8 | 32 |
| Doxycycline-resistant *S. typhi* BYG 31 | 8 | 32 |

*MIC* Minimum Inhibitory Concentration, *FLO* florfenicol.
The MICs of antibiotics were determined by the broth microdilution method (see "Methods") following the Clinical and Laboratory Standards Institute (CLSI) guidelines. The assay was done three times to confirm results (*n* = 3). Source data are provided as a Source Data file.

was a significant difference in hemoglobin (HGB) between the F8-treated group and the control group. However, there were no significant differences in the number of white blood cells (WBCs), neutrophils (Neu), lymphocytes (Lym), red blood cells (RBCs), and platelets (PLT), indicating that F8 has low toxicity to mice (Supplementary Fig. 2c–h). In TUNEL staining analysis, the immune organs (thymus, spleen, and bone marrow) treated with F8 did not show intense fluorescence compared with the positive control FLO, indicating fewer instances of cell death (Fig. 3j).

Additionally, F8 exhibited tolerance in a 14-day study of acute toxicity in mice. At a maximum dose of 5000 mg/kg, no abnormal clinical signs were observed (Fig. 3k and Supplementary Fig. 3). The cytotoxicity of F8 to mammalian cells was also low (Fig. 3l–n). Furthermore, we found that F8 has a low haemolysis rate on sheep red blood cells, showing negligible haemolysis even at 256 µM (Fig. 3o and Supplementary Fig. 4). In conclusion, these results suggest that F8 possesses good pharmacological efficacy in vivo; however, further

studies should be carried out to investigate its pharmacokinetic characters and impacts on a wider range of clinical infections.

## F8 overcomes resistance and is effective in vivo
Bacterial resistance is also an important indicator of the potential clinical application of antimicrobial drugs. To evaluate the development of F8 resistance, we conducted an in vitro study, continuously exposing *S. aureus* and *E. coli* to sub-inhibitory concentrations of F8 for 30 consecutive days. Compared with the positive control FLO, the trend of bacteria developing resistance to F8 was lower (Fig. 4a, b). Subsequently, the inhibitory effect of F8 on induced FLO-resistant *S. aureus* and FLO-resistant *E. coli* was also tested, and it was found that F8 significantly enhanced the inhibitory effect on these FLO-resistant bacteria (Fig. 4c). These results suggest that F8 holds greater advantages in avoiding bacterial resistance.

To further evaluate the in vivo antibacterial action of F8, a mouse sepsis model with FLO-resistant *S. aureus* was established. Excitingly, the survival curve showed that the survival rate of all mice infected with drug-resistant bacteria was as high as 50% within 72 h after treatment with 60 mg/kg of F8, while under the same conditions, the survival rate of the positive control FLO decreased to 0 within 48 h (Fig. 4i). This survival analysis showed that the F8 treatment group was able to significantly inhibit drug-resistant bacteria in vivo and had a significant survival advantage.

Subsequent evaluation of bacterial load measurements was consistent with the survival rate analysis. After 24 h of F8 treatment, the bacterial load in the liver, spleen, kidneys, and blood showed significant reduction (Fig. 4d–g), and furthermore, F8 exhibited a considerable inhibitory effect on the invasion of FLO-resistant *S. aureus*. Moreover, histopathological examinations revealed noticeable tissue necrosis and damage in the FLO-resistant *S. aureus* group, while F8 treatment significantly alleviated the necrosis and damage in the liver, kidney, and spleen tissues caused by bacterial infection (Fig. 4h). Overall, our results suggest that F8 effectively alleviates the severity of FLO-resistant *S. aureus* infection.

## Discovery of arcB as the antibacterial target of F8 from multi-omics analysis
To uncover the potential targets of F8 in its anti-*S. aureus* activity, we analysed the bacterial changes post-F8 treatment using transcriptomics, proteomics, and metabolomics (Fig. 5a and Supplementary Fig. 5). From transcriptomic analysis, a total of 1212 differentially expressed genes (DEGs) (598 upregulated and 614 downregulated) showed significant expression patterns before and after F8 treatment (FC < 0.5 or >2, *P* < 0.05) (Supplementary Fig. 6a). Kyoto encyclopedia

**Table 3 | MBC of broad-spectrum antibacterial small molecule compounds based on SBDD and modular synthesis**

| Organism | MBC (µM) | | | | | | | | | | | |
|---|---|---|---|---|---|---|---|---|---|---|---|---|
| | FLO | F1 | F2 | F3 | F4 | F6 | F7 | F8 | F11 | F13 | F14 | F15 |
| *E. coli* | 16 | 32 | 64 | >128 | >128 | 16 | 32 | 8 | >128 | >128 | 32 | >128 |
| *S. typhi* | 8 | 16 | 64 | >128 | >128 | 8 | 32 | 4 | >128 | >128 | 16 | >128 |
| *P. aeruginosa* | >128 | >128 | >128 | >128 | >128 | >128 | >128 | >128 | >128 | >128 | >128 | >128 |
| *S. aureus* | 32 | 64 | 64 | >128 | >128 | 32 | 64 | 16 | >128 | >128 | 64 | >128 |
| *B. subtilis* | 8 | 16 | 32 | >128 | >128 | 8 | 8 | 4 | >128 | >128 | 16 | >128 |
| *E. faecalis* | 32 | 64 | 64 | >128 | >128 | 16 | 16 | 8 | >128 | >128 | 32 | >128 |
| *P. multocida* | 8 | 16 | 8 | >128 | 128 | 4 | 8 | 4 | >128 | >128 | 4 | >128 |
| *A. pleuropneumoniae* | 32 | 64 | 64 | >128 | >128 | 8 | 64 | 8 | >128 | >128 | 32 | >128 |
| *S. suis* | 32 | 128 | 128 | >128 | >128 | 16 | 32 | 8 | >128 | >128 | 64 | >128 |
| *H. parasuis* | 64 | 128 | >128 | >128 | >128 | 32 | 64 | 16 | >128 | >128 | 64 | >128 |

*MIC* Minimum Inhibitory Concentration, *FLO* florfenicol.
The MBCs of antibiotics were determined as the lowest concentration of tested compound that killed at least 99.9% of the initial inoculums. The assay was done three times to confirm results (*n* = 3).
Source data are provided as a Source Data file.

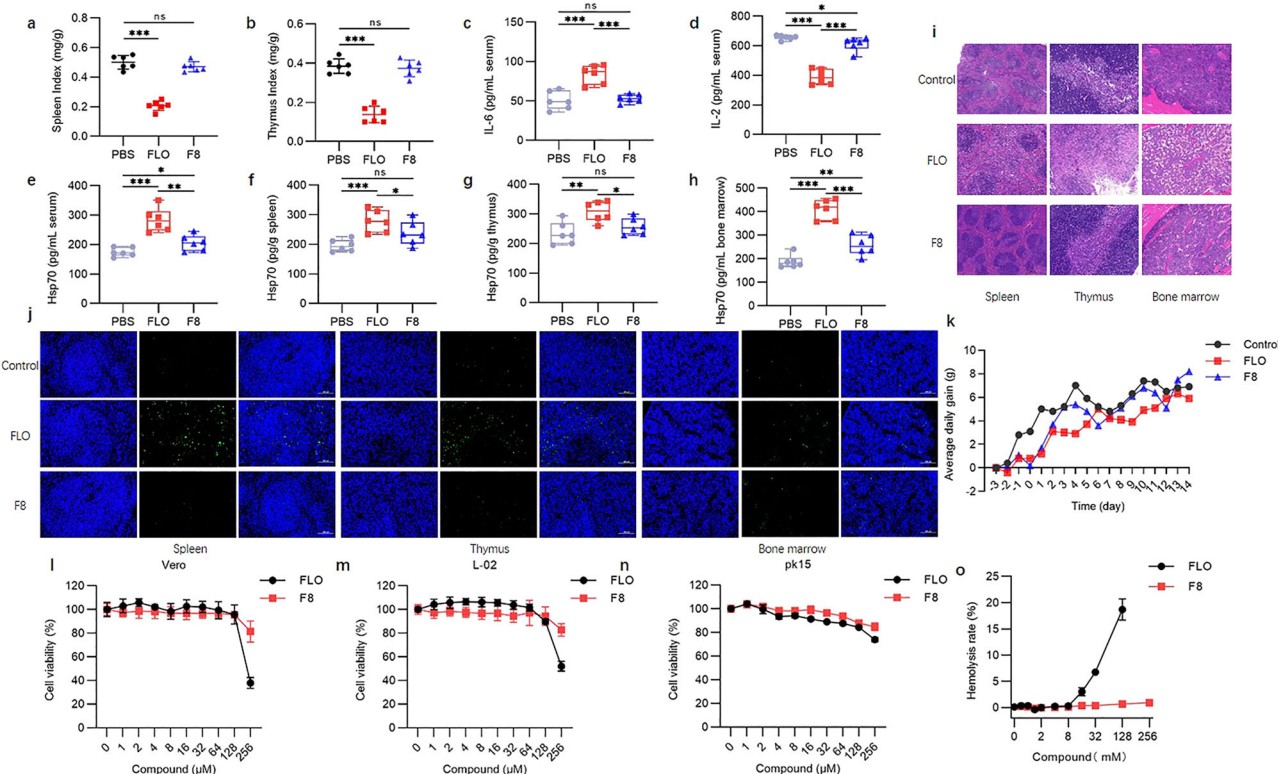

**Fig. 3 | Safety evaluation of F8. a** Spleen index. Spleen index (mg/g) = spleen weight (mg)/body weight of mice (g). (*n* = 6). (*P* < 0.0001 (FLO-PBS), *P* = 0.2194 (F8-PBS)). **b** Thymus index. Thymus index (mg/g) = thymus weight (mg)/body weight of mice (g). (*n* = 6). Statistics: two tailed *t* test. Mean with SD. (*P* < 0.0001 (FLO-PBS), *P* = 0.6396 (F8-PBS)). **c** Concentration of IL-6 in serum. (*n* = 6). (*P* = 0.0007 (FLO-PBS), *P* = 0.6481 (F8-PBS), *P* = 0.0002 (F8-FLO)). **d** Concentration of IL-2 in serum. (*n* = 6). (*P* < 0.0001 (FLO-PBS), *P* = 0.0490 (F8-PBS), *P* < 0.0001 (F8-FLO)). **e** Concentration of Hsp70 in serum. (*n* = 6). (*P* = 0.0001 (FLO-PBS), *P* = 0.0360 (F8-PBS), *P* = 0.0026 (F8-FLO)). **f**–**h** Concentration of Hsp70 in the spleen, thymus, and bone marrow. (*n* = 6). In boxplots the lower hinge represents 25% quantile, upper hinge 75% quantile, and center line the median. Statistics: two tailed *t* test. **f** (*P* = 0.0008 (FLO-PBS), *P* = 0.0467 (F8-PBS), *P* = 0.0972 (F8-FLO)). **g** (*P* = 0.0053

(FLO-PBS), *P* = 0.2509 (F8-PBS), *P* = 0.0188 (F8-FLO)). **h** (*P* < 0.0001 (FLO-PBS), *P* = 0.0085 (F8-PBS), *P* = 0.0001 (F8-FLO)). ns, not significant, \**P* < 0.05, \*\**P* < 0.01, \*\*\**P* < 0.001. **i** Histopathological assessment of the spleen, thymus, and bone marrow. Scale bar, 100 µm. **j** Apoptotic cells in the spleen, thymus, and bone marrow in different groups (TUNEL staining, under a fluorescence microscope at ×200). **k** Acute toxicity test, record of mouse body weight after a single dose of F8 or positive control FLO at 5000 mg/kg (*n* = 6). **l**–**n** Toxicity of different concentrations of F8 or positive control FLO (0–256 µM) on the Vero cells, L0-2 cells and pk15 cells (*n* = 6 biological replicates). Statistics: two tailed *t* test. Mean with SD. **o** Hemolysis rate of sheep red blood cells by different concentrations of F8 or positive control FLO (0–256 µM) (*n* = 6 biological replicates). Statistics: two tailed *t* test. Mean with SD. Source data are provided as a Source Data file.

of genes and genomes (KEGG) and gene ontology (GO) analysis revealed that pathways for cellular components (cytoplasmic, membrane, and plastid parts), molecular function (amino acid biosynthesis process, small molecule metabolic process, and oxidoreductase

activity), and energy metabolism (arginine biosynthesis, nitrogen metabolism, histidine metabolism, glycolysis, and alanine, aspartate and glutamate metabolism) were downregulated, whereas activities of RNA processing, RNA metabolic processes, and cellular macromolecule

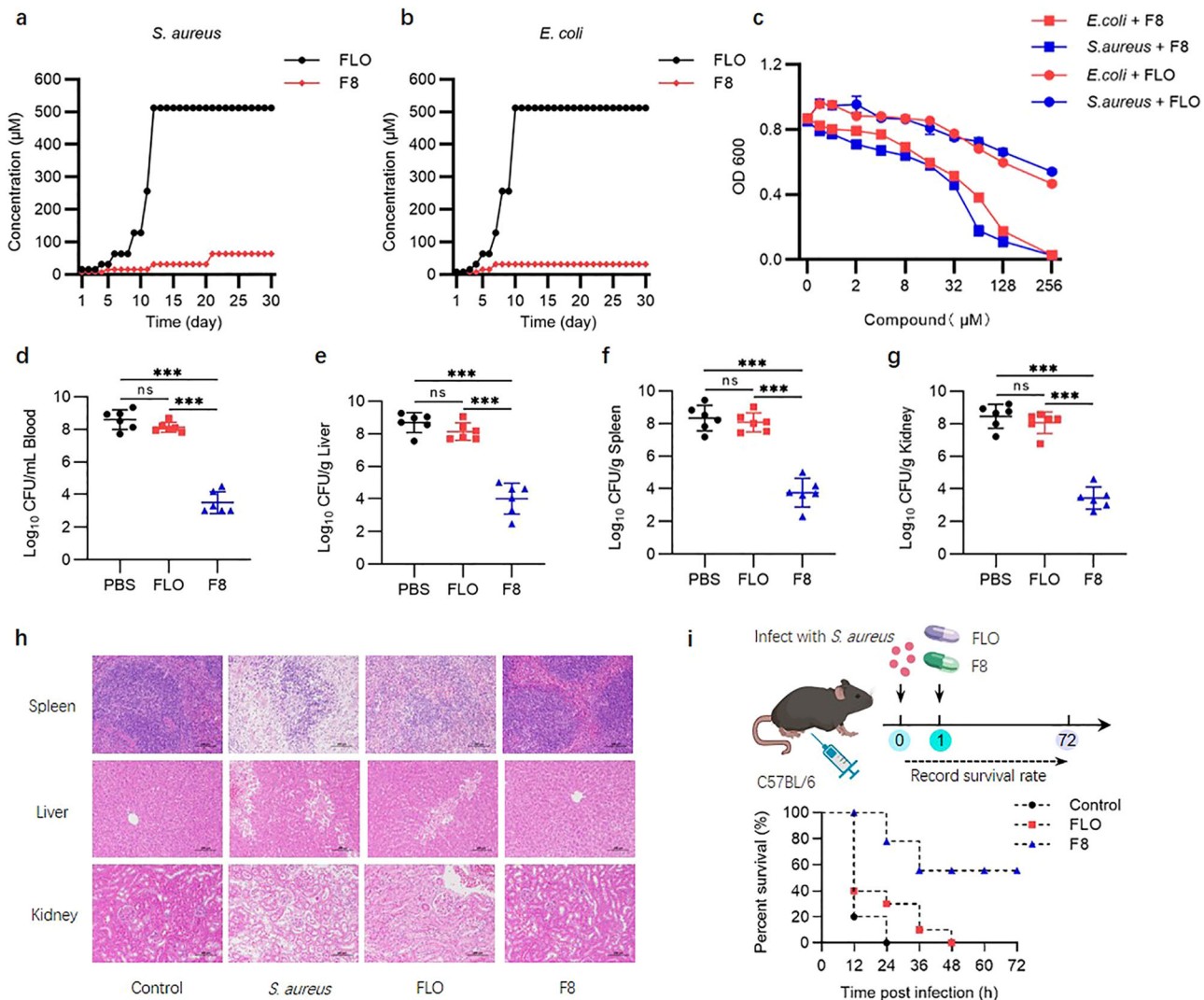

**Fig. 4 | F8 overcomes resistance and is effective in vivo. a** The development of drug resistance to F8 over 30 days of consecutive subculturing. *S. aureus* ATCC 29213 is sub-cultured daily under sub-inhibitory concentrations of F8 or positive control FLO. **b** The development of drug resistance to F8 over 30 days of consecutive subculturing. *E. coli* ATCC 25922 is sub-cultured daily under sub-inhibitory concentrations of F8 or positive control FLO. **c** Inhibition of florfenicol-resistant *S. aureus* and florfenicol-resistant *E. coli* by different concentrations of F8. ($n = 6$). Mean with SD. **d–g** Bacterial load in the blood, liver, spleen, and kidneys of different treatment groups in a mouse model of *S. aureus* bacteraemia ($n = 6$). Statistics: two tailed *t* test. Mean with SD. **d** ($P = 0.1169$ (FLO-PBS), $P < 0.0001$ (F8-PBS), $P < 0.0001$ (F8-FLO)). **e** ($P = 0.1307$ (FLO-PBS), $P < 0.0001$ (F8-PBS), $P < 0.0001$ (F8-FLO)).

**f** ($P = 0.5324$ (FLO-PBS), $P < 0.0001$ (F8-PBS), $P < 0.0001$ (F8-FLO)). **g** ($P = 0.3454$ (FLO-PBS), $P < 0.0001$ (F8-PBS), $P < 0.0001$ (F8-FLO)). ns, not significant, $*P < 0.05$, $**P < 0.01$, $***$, $p < 0.001$. **h** Histopathological assessment of the liver, spleen, and kidneys in different treatment groups in a mouse model of florfenicol-resistant *S. aureus* bacteraemia. Scale bar, 100 μm. **i** Survival curves of mice in an anti-*S. aureus* study ($n = 10$). F8/FLO were administered via oral gavage as the route of delivery. Created with BioRender.com. Source data are provided as a Source Data file.
**i** Created with BioRender.com released under a Creative Commons Attribution-NonCommercial-NoDerivs 4.0 International license (https://creativecommons.org/licenses/by-nc-nd/4.0/deed.en).

biosynthetic processes were upregulated (Supplementary Fig. 6b–f). In conclusion, these upregulated DEGs are largely associated with the metabolic disorder and degradation of secondary metabolites, whereas processes like amino acid biosynthesis, nitrogen source metabolism, and glycolysis are significantly suppressed.

Proteomic analysis revealed that after F8 treatment, there were 457 differentially expressed proteins (DEPs) (224 upregulated and 233 downregulated), with significantly downregulated genes mainly concentrated in the pathways of histidine metabolism, nitrogen metabolism, arginine biosynthesis, and alanine, aspartate, and glutamate metabolism (Supplementary Fig. 7). In the metabolomics data, there were 326 different metabolites between the control group and F8 group (Supplementary Fig. 8). KEGG pathway analysis revealed a significant downregulation in tricarboxylic acid (TCA) cycle and cofactor biosynthesis. Furthermore, after F8 treatment, molecules associated

with purine and pyrimidine metabolism, as well as ornithine, pyruvate, aspartate, glutamate, and arginine metabolism were significantly downregulated.

Subsequently, multi-omics pathway analysis was utilized to further scrutinize the data across the three omics (Fig. 5a and Supplementary Fig. 5). In the integrated transcriptomics-proteomics analysis, a strong correlation was observed in nitrogen metabolism, amino acid synthesis, and the stress response. Similarly, in the integrated transcriptomics-metabolomics analysis, amino acid synthesis, nitrogen metabolism, and energy metabolism were significantly downregulated. After conducting KEGG enrichment analysis of significantly downregulated proteins and metabolites, we noticed a significant downregulation in arginine metabolism, alanine, aspartate, and glutamate metabolism. Finally, multi-omics analysis suggested that F8 down-regulates the biosynthesis of nitrogen metabolism, arginine

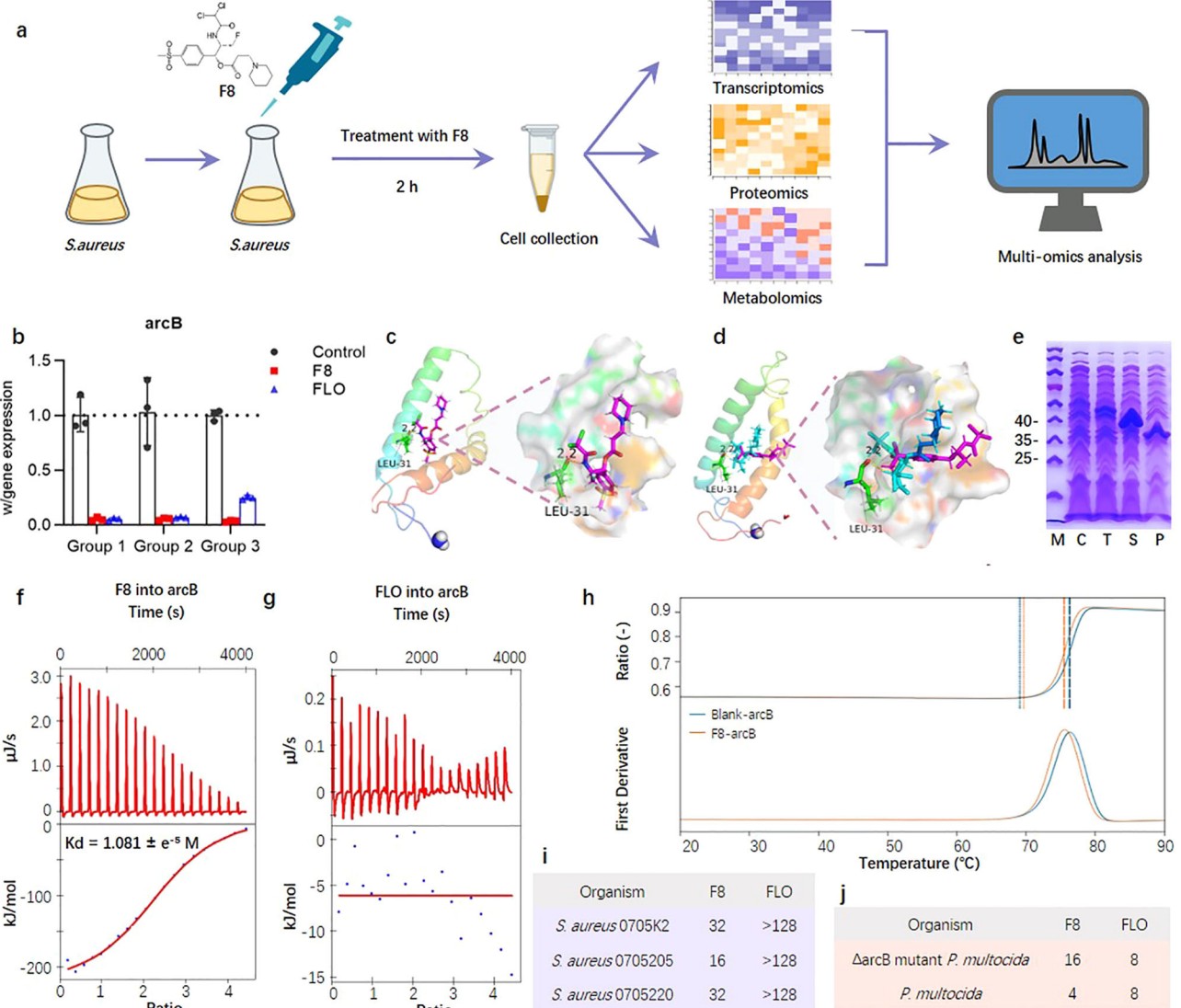

**Fig. 5 | Binding of F8 with arcB. a** Experimental schematic for the multi-omics experiment. Created with BioRender.com. **b** Gene-level differences of arcB in *S. aureus* treated with F8 or positive control FLO. Groups 1–3 represent three sets of transcriptomic samples, respectively. (*n* = 3). Mean with SD. **c** Simulated image of the F8 4 Å DBD in arcB (PDB:2ksd) generated by PyMOL software, with residues LEU-31 shown in green. **d** The superposition of the crystal structure of the DBDs of F8 and FLO bound to arcB, with residues LEU-31 shown in green, FLO in light purple, and F8 in blue. **e** The expression of arcB was analysed by SDS-PAGE, and compared with the negative control (C, empty pET28a vector). The target protein was expressed in the form of total protein (T), soluble (S), and insoluble (P) of the lysate. The expected size of arcB is 37.74 kDa. Three times each experiment was repeated independently with similar results. **f, g** ITC detection of ligand target-binding affinity. **h** Detection of thermal displacement ($T_m$ change) by DSF. **i** MIC of F8 against strains with constitutive cfr resistance. **j** MIC of F8 against the arcB deletion mutant. The assay was done three times to confirm results (*n* = 3). FLO florfenicol. Source data are provided as a Source Data file. **a** Created with BioRender.com released under a Creative Commons Attribution-NonCommercial-NoDerivs 4.0 International license (https://creativecommons.org/licenses/by-nc-nd/4.0/deed.en).

biosynthesis, and alanine, aspartate, and glutamate metabolism. Arginine metabolism in bacteria is tightly regulated, and arginine is crucial for cell growth and biomembrane formation[21], thus, we infer that F8 exerts its antibacterial effects through multifaceted pathways including participation in the arginine degradation metabolic pathway, bacterial cell membranes, and energy metabolism.

Based on the multi-omics analysis, downregulation of nitrogen metabolism and arginine biosynthesis was observed in all three omics datasets, hence, we hypothesize that an important antibacterial target of F8 against *S. aureus* could potentially involve the arginine degradation metabolic pathway. Within the arginine degradation metabolic pathway, arginine deiminase (arcA), arcB, and carbamate kinase (arcC) are categorized as an arginine deiminase pathway (ADI)[22]. The RT-PCR results are consistent with the transcriptomics data (Fig. 5b, Supplementary Fig. 9a–i and Supplementary Table 3). Notably, since arcB is involved in

arginine degradation and is also indispensable for arginine biosynthesis, studies have shown that arcB levels might limit arginine biosynthesis of *S. gordonii*[21]. Expression of arcB is also known to be affected following treatment with antibiotics, including gentamicin and polymyxin B[23,24]. Therefore, we preliminarily infer arcB as a possible target of F8.

## ArcB binding by F8

Given that F8 significantly impacts the expression of arcB in multi-omics analysis, we posit that F8 exerts its antibacterial activity by targeting the arcB protein and affecting its function. To further detail the binding mode of F8 and arcB, we searched for the crystal structure of the arcB protein (PDB: 2ksd) in the PDB and performed molecular docking using the SYBYL-X software. We found that F8 is capable of binding with the arcB protein and occupying the binding pocket (Fig. 5c, d and Supplementary Fig. 10), and that the cyclic structure of

F8 nestles into the cavity of the receptor protein, enhancing the tightness of the binding. As shown in Fig. 5c, F8 binds directly to arcB through hydrogen bonding interactions with LEU-31 of arcB.

To further validate that F8 can directly bind to arcB, we expressed the arcB protein and then examined their interactions (Fig. 5e and Supplementary Fig. 11a). Isothermal titration calorimetry (ITC) measurements further confirmed the predicted interaction between F8 and the active pocket of arcB. The results showed that F8 directly and specifically binds to the arcB protein (Kd = $1.081 \pm e^{-5}$ M) (Fig. 5f, g and Supplementary Fig. 11b, c). Subsequently, differential scanning fluorimetry (DSF) was employed to assess protein thermal stability induced by compound binding to evaluate the interaction. A thermal shift ($\Delta T_m$ change) was observed when the arcB protein was incubated with F8 compared to the blank control (Fig. 5h), providing evidence for the binding of F8 to arcB protein. These results collectively demonstrate that F8 can directly bind to arcB with high affinity in vitro.

To further confirm that arcB is an important factor in the increase of F8 activity, we selected three bacterial strains with constitutive cfr resistance for MIC experiments. The results showed that F8 still maintained significant activity in these strains with cfr resistance, with MIC values of 16–32 μM, while the activity of FLO was significantly reduced, with MIC values >128 μM (Fig. 5i). This indicates that the antimicrobial mechanism of F8 does not depend entirely on ribosome inhibition and that arcB is an important target in the antimicrobial activity of F8. Furthermore, we examined the antimicrobial activity of F8 against the arcB deletion mutant. The results showed that F8 has an attenuated antimicrobial effect in the ΔarcB mutant *P. multocida* with a MIC of 16 μM, whereas FLO has an unchanged MIC of 8 μM (Fig. 5j), further suggesting that arcB is an effective target for F8.

## Bactericidal mechanism of F8

Given the wide-ranging regulatory ability of arcB, which is crucial for cell growth and biofilm formation, we hypothesize that F8 may exert its antibacterial effect through bacterial cell membrane damage. To this end, we examined the bacterial extracellular contents, showing an increasing trend in the concentration of DNA, RNA, and protein at various time intervals after F8 treatment at different concentrations (4×MIC, 2×MIC, and MIC) (Fig. 6a, b and Supplementary Fig. 12). The reason for this observation may be that F8 can alter bacterial cell membrane permeability, resulting in an increased efflux of these three substances. Alternatively, F8 might have a destructive effect on the bacterial cell membrane structure, resulting in a direct efflux of bacterial contents. Regardless of the reason, the reduction in bacterial DNA, RNA, and protein content can disrupt its normal physiological activities, leading to bacteriostatic or bactericidal effects. Further examination of bacterial cytoplasmic membrane permeability by PI staining showed that F8 induced increased membrane permeability in *S. aureus* (Fig. 6c, d and Supplementary Fig. 13). This demonstrates that F8 can alter bacterial cell membrane permeability or disrupt cell membrane structure, subsequently exerting bacteriostatic or bactericidal effects.

Consistent with this, changes in membrane rigidity disrupted the bacterial homeostasis, leading to fundamental metabolic disarray, including the dissipation of proton motive force (PMF)[25]. Thus, we evaluated the ΔpH, a key component of PMF, in *S. aureus* using the fluorescent probe BCECF-AM[26]. The experiment was conducted by treating the bacteria with 4×MIC of F8. Within 1 h after the medication was added, the ΔpH significantly dissipated (Fig. 6e, f and Supplementary Fig. 14), which is consistent with the bactericidal effect. This may indicate that changes in bacterial membrane rigidity disrupt the bacterial steady state, causing fundamental metabolic disarray and leading to the dissipation of proton motive force.

On the other hand, disruption of membrane homeostasis often contributes to the accumulation of reactive oxygen species (ROS)[27]. Moreover, previous studies have shown that the deletion of arcB in *E. coli* leads to an increased sensitivity to hydrogen peroxide under aerobic conditions[28]. Our investigations revealed that F8 increased intracellular ROS content (Fig. 6g, h and Supplementary Fig. 15), which may correspondingly exacerbate membrane damage and further disrupt bacterial homeostasis. Endogenous ROS plays a crucial role in bactericidal activity[29]. F8 is expected to stimulate the production of ROS in various ways, while also inhibiting the activity of arcB, leading to a decrease in antioxidant capacity. This dual approach may lead to more severe oxidative damage, thereby achieving better bactericidal effects and potentially avoiding the resistance effect caused by a single factor. Interestingly, we observed that ATP levels increased in a dose-dependent manner following F8 treatment (Fig. 6i, j and Supplementary Fig. 16). Such results are in line with previous research, suggesting that bactericidal antibiotics are associated with accelerated respiration[30]. These findings provide convincing phenotypic support for the notion that F8 disrupts bacterial cell membranes and induces a certain degree of oxidative damage.

## Discussion

Antimicrobial drug resistance has evolved into a global health crisis. Antibiotics have been indispensable in the battle against bacteria, saving millions of lives[31–34]; however, the overuse of antibiotics has led to a rapid increase in the number and types of resistant bacteria, particularly the emergence of superbugs, making it crucial to develop new and more effective strategies to treat bacterial infections[5,35,36]. Nowadays, the development of antibiotics is constantly challenged, resulting in few new antibiotics being introduced into clinical practice[14]. Despite considerable efforts, almost no new antibiotics have been approved for clinical use in recent years, therefore, there is an urgent need for feasible approaches to increase the number of therapeutic options.

Based on the SBDD and modular synthesis, we determined that F8 is the optimal modifying compound. It further enhanced its antibacterial effect against pathogenic bacteria such as *E. coli*, *S. aureus*, *S. typhi*, *P. multocida*, and *H. parasuis*, and imparted measurable activity against the particularly challenging *P. aeruginosa*. The most notorious drug-resistant bacterium, MRSA, exhibits resistance to a multitude of antibiotics and is spreading at an alarming worldwide. Currently, there is a significant lack of antibiotics capable of combating MRSA[37]. Our research findings indicate that F8 has a strong inhibitory effect on MRSA, with a MIC value of 8 μM, consistent with the results observed in the standard strains. Polymyxins, particularly polymyxin B, have become the last line of therapy against multi-drug-resistant Gram-negative bacteria[38]. However, the rapid development of resistance has led to the emergence of more and more polymyxin B-resistant bacteria. The MIC value of F8 against polymyxin B-resistant *E. hormaechei* is 16 μM, which is a promising lead compound for resisting polymyxin B resistance. Sulfamethoxazole is reported to be one of the most widely used sulfonamide antibiotics in the world; however, the increasingly drug resistance greatly limits its utility[39]. The MIC value of F8 against sulfamethoxazole-resistant *S. typhi* only 4 μM. In addition, F8 also showed bactericidal effects on FLO-resistant *S. suis*, FLO-resistant *H. parasuis* and doxycycline-resistant *S. typhi*.

In this study, we chose structural modification to perform at the β-hydroxy position of the bioactive scaffold. The hydroxyl group offers significant advantages for the fusion of small molecules, enabling the generation of structures with straightforward synthetic routes and conforming to easily-synthesizable mechanisms. The increase in molecular weight is a major limitation. In fact, the permeability barriers caused by a high molecular weight are the primary reasons for the limited antibacterial activity exhibited by most antibiotics[40]. Given the bacterial cell wall and membrane structure, antibacterial compounds with a high molecular weight (>600 g/mol) do not pass through non-selective protein channels, restricting uptake by bacteria to receptor-mediated endocytosis or passive diffusion[40,41]. In this study, we have

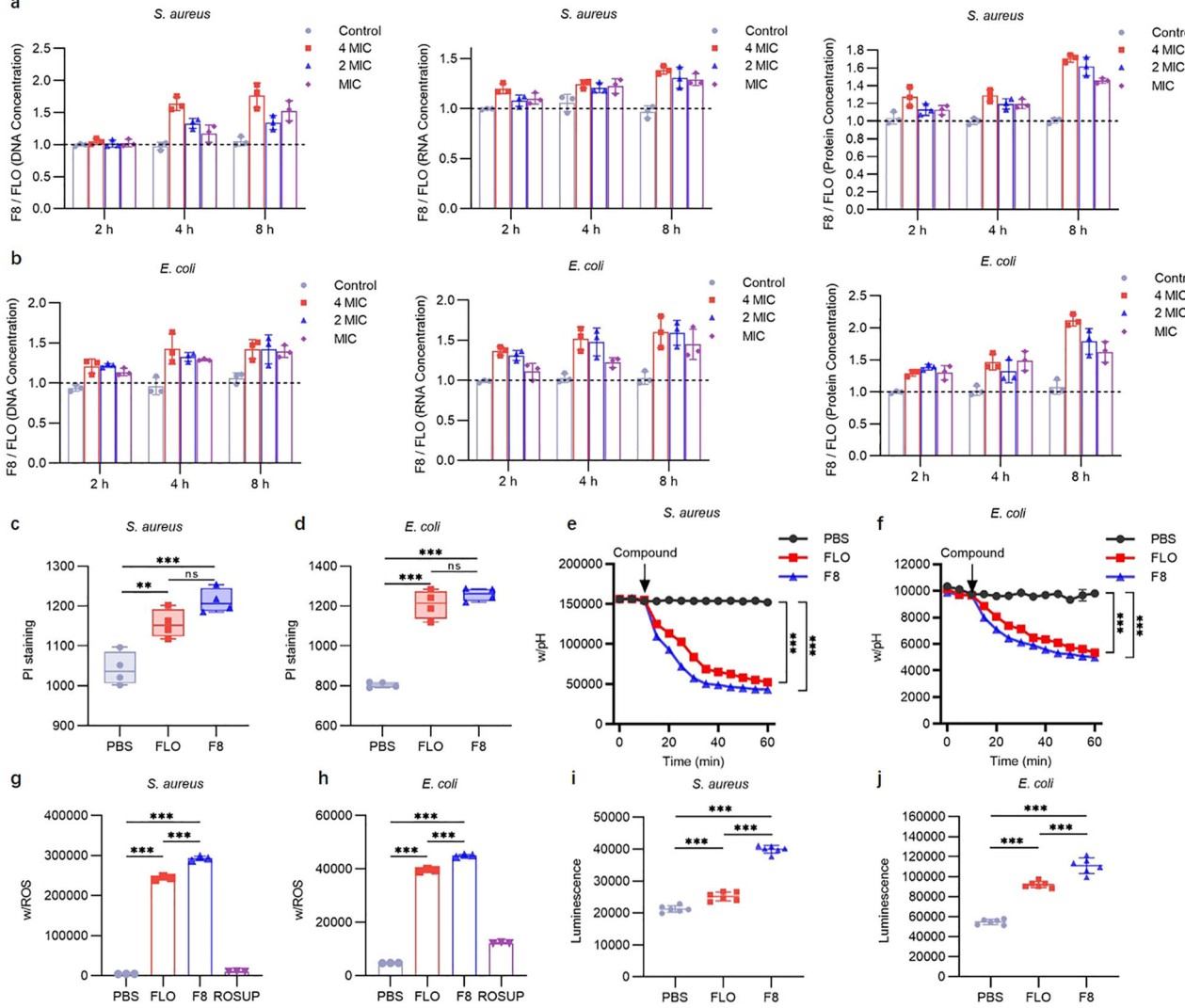

**Fig. 6 | Bactericidal mechanism of F8. a**, **b** Concentrations of extracellular DNA, RNA, and protein after treatment with F8 in *S. aureus* and *E. coli* (*n* = 3). Mean with SD. **c**, **d** Increased membrane permeability in *S. aureus* and *E. coli* after treatment with 4×MIC of F8. Membrane permeability measured by propidium iodide (PI), with excitation/emission wavelengths at 535 nm/615 nm (*n* = 4). Statistics: two tailed *t* test. In boxplots the lower hinge represents 25% quantile, upper hinge 75% quantile, and center line the median. **c** (*P* = 0.0062 (FLO-PBS), *P* = 0.0006 (F8-PBS), *P* = 0.0517 (F8-FLO)). **d** (*P* < 0.0001 (FLO-PBS), *P* < 0.0001 (F8-PBS), *P* = 0.2512 (F8-FLO)). **e**, **f** Dissipation of ΔpH by F8 and FLO in *S. aureus* and *E. coli*. Exponential-phase *S. aureus* and *E. coli* were incubated with pH fluorescence probe BCECF-AM. The cells were treated with 4×MIC of F8 and FLO, and the intracellular pH was determined. Fluorescence measurements were taken at 5-min intervals using an excitation/emission wavelength of 488 nm/535 nm. (*n* = 3). Statistics: two tailed *t* test. **e** (*P* = 0.0002 (FLO-PBS), *P* = 0.0001 (F8-PBS)). **f** (*P* = 0.0002 (FLO-PBS), *P* < 0.0001 (F8-PBS)). **g**, **h** Accumulation of intracellular ROS in *S. aureus* and *E. coli* after treatment with 4×MIC of F8 and FLO. The fluorescence probe DCFH-DA (10 μmol/L) was added to the bacterial suspension. The cells were treated with 4×MIC of F8 and FLO. After incubation for 2 h, the fluorescent values were measured with an excitation/emission wavelength of 488 nm/525 nm. ROSUP was used as the positive control. (*n* = 3). Statistics: two tailed *t* test. **g** (*P* < 0.0001 (FLO-PBS), *P* < 0.0001 (F8-PBS), *P* = 0.0002 (F8-FLO)). **h** (*P* < 0.0001 (FLO-PBS), *P* < 0.0001 (F8-PBS), *P* = 0.0001 (F8-FLO)). **i**, **j** Changes in intracellular ATP levels in *S. aureus* and *E. coli* after treatment with F8 and FLO. After treatment with 4×MIC of FLO or F8 for 2 h, bacterial cells were lysed using lysozyme, and following centrifugation, the supernatant was collected to measure the intracellular ATP levels. Luminescence was measured using the Infinite M200 Microplate reader (Tecan). (*n* = 6). Statistics: two tailed *t* test. **i** (*P* = 0.0002 (FLO-PBS), *P* < 0.0001 (F8-PBS), *P* < 0.0001 (F8-FLO)). **j** (*P* < 0.0001 (FLO-PBS), *P* < 0.0001 (F8-PBS), *P* = 0.0003 (F8-FLO)). Source data are provided as a Source Data file.

selected the simple, low-molecular-weight, and easily-hybridisable FLO as the primary bioactive scaffold. Targeting the key bacterial ribosome 50 S subunit PTC region of FLO, we employed a computer-aided drug design to develop a modular synthetic route, and investigated a broad range of structural variations. We discovered that, in general, short fragments have higher activity compared to their homologs. Smaller spatial conformations within the binding pocket reduce the level of inherent self-collision incidences the ligand may experience, leading to increased stability in binding. This also aligns with the lower molecular weight required for cell permeability. Although the conceptualization of an ideal blueprint may seem trivial, the discovery of F8 undoubtedly

reaffirms that chemical synthesis remain powerful means to supplement our arsenal of antibiotics.

F8 slows down the development of drug resistance and exhibits significant in vitro and in vivo antibacterial effects on resistant bacteria, which means that compared to traditional antibiotics, F8 has different mechanisms of action or exerts its antibacterial effect through the combination of several mechanisms of action. Multi-omics analysis indicates that F8 affects multiple metabolic routes, including nitrogen metabolism, arginine biosynthesis, and alanine, aspartic acid, and glutamate biosynthesis. Arginine metabolism is strictly regulated in bacteria and arginine is essential for cell growth and biofilm

formation[31]. In the arginine catabolism pathway, arcA, arcB, and arcC are classified as the arginine deiminase pathway[22]. Research indicates that in *S. aureus*, the activation of the arginine deiminase pathway confers resistance to vancomycin[42], which can explain why *S. aureus* showed a lower resistance to F8 in the process of continuous passage. Meanwhile, in ITC and DSF assays, we discovered and validated the target protein arcB of F8. This finding is consistent with our multi-omics analysis.

As ArcB is involved in arginine degradation metabolism biosynthesis[43–46], the report by Jakubovics et al.[21] suggests that arcB levels may limit the biosynthesis of arginine in *S. gordonii*. Our research results also indicate that F8 can alter bacterial cell membrane permeability or disrupt the cell membrane structure, thus exerting an antibacterial or bactericidal effect. The expression of arcB is also known to be affected after treatment with antibiotics, including gentamicin and polymyxin B, suggesting that cell membrane stress influences ArcB activity[23,24]. In addition, our research indicates that F8 increases the content of intracellular ROS, which may correspondingly exacerbate membrane damage and further disrupt bacterial homeostasis. Several studies have determined that arcA/arcB play a role in this ROS resistance, and in fact, the deletion of arcA or arcB has been shown to lead to increased sensitivity to hydrogen peroxide under aerobic conditions in *E. coli*[28]. It has been demonstrated that the arcA/arcB can regulate the synthesis and transport of proteins and amino acids, thereby affecting the adaptability of *E. coli* under ROS stress[28], thus, it is hypothesized that bactericidal antibiotics lead to metabolic instability and toxic ROS formation as part of their lethal action[27,30,47–49]. Importantly, mutants of arcB may affect bacterial growth[50]. We further examined the antibacterial activity of F8 against arcB deletion mutants and bacterial strains with constitutive cfr resistance. Our findings indicate that F8 retains significant activity in strains with constitutive cfr resistance and that its antimicrobial mechanism differs from that of FLO and does not depend exclusively on ribosome inhibition. In addition, the antimicrobial effect of F8 was attenuated in the arcB deletion mutant. Therefore, arcB is an important target in the antimicrobial activity of F8.

Given that F8 exhibits significant antimicrobial activity and has the potential to effectively mitigate the development of resistance, further extensive work is needed to expand its antibacterial spectrum against both standard and resistant strains, as well as to assess its efficacy in diverse biological environments. Our study suggests that F8 targets the arcB gene, which may result in different mechanisms of antibacterial action. However, we do not rule out the possibility that F8 may also affect other biological pathways and molecular targets. For example, it is important to determine whether F8 is susceptible to resistance mechanisms associated with phenicols in clinical settings, as this could influence the development of resistance. Our study preliminarily investigated the safety profile of F8, encompassing acute toxicity, organ damage, immunotoxicity, and hematoxicity. Future research will need to delve deeper and broaden the scope of investigation into its side effects, including long-term toxicity, genotoxicity, and other potential adverse effects. In addition, current environmental contamination from antibiotics is a matter of concern. It is particularly crucial to conduct an environmental risk assessment for F8, which should include its degradability in aquatic and soil environments, the assessment of health risks to the biotic community, and the potential transfer of resistance genes.

In summary, our research findings suggest that F8 is a broad-spectrum antibacterial small molecule compound, exhibiting significant in vivo and in vitro antibacterial activity against both standard and drug-resistant strains, with a low propensity for the development of resistance. This mechanism of antibacterial action is thought to be based on competitive binding with arcB, thereby activating multiple pathways. Together, our results suggest that F8 may be a promising candidate drug for developing antibiotic formulations to combat antibiotic-resistant bacteria-associated infections.

## Methods

All research complies with the relevant ethical regulations. Study protocols were approved by the Animal Welfare and Ethics Committee of Huazhong Agricultural University (Wuhan, China) for all animal experiments.

### Strains, media, and growth conditions

*E. coli* (ATCC 25922), *S. aureus* (ATCC 29213), *S. typhi* (CICC 110420), *P. aeruginosa* (ATCC 9027), *B. subtilis* (CMCC 63501), *E. faecalis* (ATCC 29212), *P. multocida* (ATCC 43137), *A. pleuropneumoniae* (ATCC 27088), *S. suis* (CVCC 606), *H. parasuis* (ATCC 19417), methicillin-resistant *S. aureus* (B1-1), polymyxin B-resistant *E. hormaechei* (wb 4), FLO-resistant *S. suis* (1136, 1194, 1197, 1655, 1658, 1669), FLO-resistant *H. parasuis* (1565, 1614, 1651), ampicillin-resistant *S. typhi* (BYG 9), sulfamethoxazole-resistant *S. typhi* (BYG 21) and doxycycline-resistant *S. typhi* (BYG 11, BYG 25, BYG 31), strains with constitutive cfr resistance (*S. aureus* 0705K2, *S. aureus* 0705205, *S. aureus* 0705220) (provided by Shaowen Li from Huazhong Agricultural University), ΔarcB mutant *P. multocida* (provided by Bin Wu from Huazhong Agricultural University)[51] were used in this study. Trypticase soy broth (TSB), Mueller-Hinton broth (MHB), Muller-Hinton agar (MHA), LB, and LB nutrient agar were bought from Haibo Biotechnology (Qingdao, China). Cells were grown at 37 °C on a rotating shaker at 300 rpm in flasks, or at 900 rpm in plate shakers.

### Molecular docking studies

Preparation of the binding site: The X-Ray crystal structure of the 50 S subunit PTC during translation termination (the PTC region) with the PDB code 6c4h was used for the virtual screening. The binding pockets were confirmed using SYBYL-X Suite with default parameters. Hydrogen atoms were added to the structure, and considerations were made regarding correct orientation of Asn and Gln sidechains, ligands, and protein charges. All waters were removed except for the iron/sulphur complex. A diverse subset of the compound library (i.e., SPECS with 1,084,348 small-molecule compounds and a Topscience compound library with 16,594 natural products) was prepared to generate three-dimensional configurations by Surflex for searching SYBYL-X with all options set as default. Docking projects were conducted using Surflex-Dock GeomX in the SYBYL-X Suite. Scoring calculations were calculated by CScore in the SYBYL-X Suite.

### Design and synthesis of broad-spectrum antibacterial small molecule compounds

The synthesis work was performed as follows: N,N-dimethylformamide (DMF, 20 mL) was added to a round-bottom flask, followed by the addition of 3-(1-Piperidinyl)propanoic acid (CAS: 26371-07-3, 2 g) until complete dissolution. Subsequently, 4-dimethylaminopyridine (DMAP, 1.5 g), 1-(3-dimethylaminopropyl)-3-ethyl carbodiimide hydrochloride (EDC, 3 g), and FLO (2,2-Dichloro-N-[(1 R,2 S)-3-fluoro-1-hydroxy-1-(4-methylsulfonylphenyl)propan-2-yl]acetamide) (3 g) (Longxiang, China) were sequentially added. The reaction mixture was stirred at 25 °C for 24 h. The output was monitored by thin-layer chromatography (TLC) (hexyl hydride/ethyl acetate [hexane/EtOAc] = 1/1, retention factor [Rf] = 0.25) and visualized by a 5% vanillin sulfuric acid/ethanol solution. The product was purified by column chromatography on a silica gel (hexane/EtOAc=2/1). Other derivatives, such as F1, F2, F3, F4, F6, F7, F11, F13, F14, and F15, were synthesized using the same procedure. Broad-spectrum antibacterial small molecule compounds were characterized by various techniques, including MS, $^1$H-NMR, and $^{13}$C-NMR (Fig. 1c, Supplementary Table 2 and Supplementary Notes).

### In vitro susceptibility

The MICs of antibiotics were determined by the broth microdilution method following the Clinical and Laboratory Standards Institute (CLSI) guidelines. Briefly, single bacterial colonies were cultured in

MHB at 37 °C at 220 rpm for 8–12 h. Subsequently, the potential lead candidates or other antibiotics were diluted two-fold in MHB and mixed with an equal volume of bacterial suspensions in MHB containing approximately $1 \times 10^6$ CFU/mL in a clear UV-sterilized 96-well microtiter plate. The plate was placed in the incubator for 18 h at 37 °C, then the MIC values were read. MIC values were defined as the lowest concentrations of antibiotics with no visible growth of bacteria. The MBC values were identified as the lowest drug concentration to kill over 99.9% of bacteria. An aliquot of 10 μL bacterial suspension from each well was plated onto MH agar plates. After the plates were incubated at 37 °C for 24 h, MBC values were determined by visual inspection of CFU on the agar.

## Growth curves of bacteria

A single colony of *E. coli* ATCC 25922 or *S. aureus* ATCC 29213 was selected and inoculated into MHB for cultivation. The cultures were incubated at 37 °C with shaking at 200 rpm for overnight growth. The overnight culture was standardized to a 0.5 McFarland turbidity standard. It was then diluted at 1:100 in MHB, and adjusted to approximately $1 \times 10^6$ CFU/mL. Different concentrations of FLO or F8 were added to a 96-well microplate and mixed with an equal volume of bacterial dilution, then sealed with a plate lid. The plate was incubated at 37 °C, and the wavelength of 600 nm was measured every 1 h using an Infinite M200 Microplate reader (Tecan).

## Time-kill curves

A single colony of *E. coli* ATCC 25922 or *S. aureus* ATCC 29213 was selected and inoculated into MHB for cultivation. The cultures were incubated at 37 °C with shaking at 200 rpm for overnight growth. The overnight cultured bacteria were diluted at 1:100 in MHB and adjusted to approximately $1 \times 10^6$ CFU/mL. Different concentrations of FLO and F8 (4×MIC, 2×MIC, MIC, 1/2×MIC) were added to the diluted bacterial suspension. The cultures were then incubated at 37 °C with shaking at 200 rpm. After incubation for 30 min, 1, 2, 4, 8, and 24 h, 100 μL aliquots were taken, serially diluted 10-fold, then inoculated on MHA plates (with the inclusion of three parallel plates), and the colony forming units (CFU) were calculated after incubation for 24 h at 37 °C.

## Membrane integrity assay

Overnight cultures of the bacterial fluid were centrifuged and resuspended in 0.01 mol/L PBS (pH 7.4), and the bacterial suspensions were adjusted to an $OD_{600}$ nm of approximately 0.5. Subsequently, FLO or F8 was added at a final concentration of 4×MIC, and the bacterial suspensions were incubated at 37 °C for 2 h without light. The suspensions were then incubated with propidium iodide (PI, Thermo Scientific, P1304MP) at a final concentration of 10 nmol/L for 20 min before the fluorescence values were quantified using an Infinite M200 Microplate reader (Tecan) at an excitation wavelength of 535 nm and an emission wavelength of 615 nm.

## Membrane permeability measurement

Overnight cultures of the bacterial fluid were centrifuged and resuspended in 0.01 mol/L PBS (pH 7.4), and the bacterial suspensions were adjusted to approximately $1 \times 10^6$ CFU/mL. The diluted bacterial cultures were supplemented with different concentrations of FLO and F8 (4×MIC, 2×MIC, MIC) and incubated at 37 °C with shaking at 150 rpm. After incubation for 2, 4, and 8 h, the bacterial suspensions were collected, and the supernatant was obtained by centrifugation at 12,000 rpm for 2 min. The DNA, RNA, and protein contents were measured using a Nano Drop microspectrophotometer.

## ΔpH measurement

Overnight cultures of the bacterial fluid were centrifuged and resuspended in 0.01 mol/L PBS (pH 7.4), and the bacterial suspensions were

adjusted to an OD600 nm of approximately 0.5. Subsequently, 10 μL of FLO or F8 (final concentration of 4×MIC) was added, followed by the addition of a pH-sensitive fluorescent probe, BCECF-AM, at a final concentration of 10 μmol/L. The mixture was then incubated at 37 °C and fluorescence measurements were taken every 5 min using the Infinite M200 Microplate reader (Tecan) with an excitation wavelength of 488 nm and an emission wavelength of 535 nm.

## ATP determination

Extracellular and intracellular ATP levels were determined using the Enhanced ATP Assay Kit (Beyotime, Cat. No. S0027). Overnight cultures of the bacterial fluid were centrifuged and resuspended in 0.01 mol/L PBS (pH 7.4), and the bacterial suspensions were adjusted to an OD600 nm of approximately 0.5. After treatment with 4×MIC of FLO or F8 for 2 h, the bacterial cultures were centrifuged at 12,000 rpm for 5 min at 4 °C, and the supernatant was collected to determine extracellular ATP levels. Meanwhile, the bacterial precipitate was lysed with lysozyme, centrifuged, and the supernatant was prepared to determine the intracellular ATP levels. The assay solution was added to a 96-well plate and incubated for another 5 min at room temperature. The supernatants were added to the well and mixed quickly before recording in the model of luminescence using the Infinite M200 Microplate reader (Tecan).

## ROS measurement

Overnight cultures of the bacterial fluid were centrifuged and resuspended in 0.01 mol/L PBS (pH 7.4), and the bacterial suspensions were adjusted to an OD600 nm of approximately 0.5. The fluorescent probe, 2′,7′-dichlorofluorescein diacetate (DCFH-DA) (10 μmol/L), was used to detect ROS accumulation in the bacterial fluids after FLO or F8 treatment, following the manufacturer's instructions (Beyotime, Cat. No. S0033). Briefly, DCFH-DA was added to the bacterial suspension and incubated at 37 °C for 20 min. After washing three times with 0.01 mol/L PBS, 190 μL of the bacterial suspension was added to a 96-well microplate and mixed with 10 μL of FLO or F8 (final concentration of 4×MIC). After incubation for 2 h, the fluorescent values were measured using the Infinite M200 Microplate reader (Tecan) with an excitation wavelength of 488 nm and an emission wavelength of 525 nm.

## Transcriptomics

Untreated and FLO- or F8-treated *S. aureus* ATCC 29213 were used for transcriptomics studies. FLO or F8 (final concentration of 4×MIC) was added to the bacterial suspension and incubated for 2 h. The bacterial cultures were then washed three times with 0.01 mol/L PBS (pH 7.4) and centrifuged at 8000 rpm for 10 min at 4 °C to collect the bacterial precipitates. RNA-Seq was performed using the Illumina platform at Shanghai Majorbio Bio-pharm Technology Co., Ltd. (Shanghai, China). Total RNA was extracted from the samples, and the concentration and purity of the extracted RNA were determined using Nanodrop 2000. The integrity of the RNA was assessed by agarose gel electrophoresis, and the RNA integrity was further evaluated using the Agilent 2100 Bioanalyzer. Genes with adjusted $P < 0.05$, identified by DESeq, were considered differentially expressed. According to the KEGG analysis, differentially-significant genes were assigned to different functional groups. The data was uploaded to the Majorbio cloud platform (https://cloud.majorbio.com) after the database search for data analysis.

## Metabolomics

Untreated and FLO- or F8-treated *S. aureus* ATCC 29213 were used for metabolomics studies. FLO or F8 (final concentration of 4×MIC) was added to the bacterial suspension and incubated for 2 h. The bacterial cultures were then washed three times with 0.01 mol/L PBS (pH 7.4) and centrifuged at 8000 rpm for 10 min at 4 °C to collect the bacterial precipitates. In total, 50 mg bacterial precipitates were accurately weighed,

and the metabolites were extracted using a 400 μL methanol:water (4:1, v/v) solution with 0.02 mg/mL L-2-chlorophenylalanin as an internal standard. The mixture was allowed to settle at −10 °C, and was treated using the high throughput tissue crusher, Wonbio-96c (Shanghai Wanbo Biotechnology Co., LTD), at 50 Hz for 6 min, followed by sonication at 40 kHz for 30 min at 5 °C. The samples were placed at −20 °C for 30 min to precipitate the proteins. After centrifugation at 13,000× $g$ at 4 °C for 15 min, the supernatant was carefully transferred to sample vials for LC-MS/MS analysis. The instrument platform for this LC-MS analysis was UHPLC-Q Exactive HF-X system of Thermo Fisher Scientific. The data was uploaded to the Majorbio cloud platform (https://cloud.majorbio.com) after the database search for data analysis.

## Proteomics

Untreated and FLO- or F8-treated *S. aureus* ATCC 29213 were used for proteomics studies. FLO or F8 (final concentration of 4×MIC) was added to the bacterial suspension and incubated for 2 h. The bacterial cultures were then washed three times with 0.01 mol/L PBS (pH 7.4) and centrifuged at 8000 rpm for 10 min at 4 °C to collect the bacterial precipitates. The bacterial precipitates were removed and put on ice. An appropriate amount of protein lysate was then added, (8 M urea, 1% SDS) and the precipitates were sonicated for 2 min at a low temperature, then split for 30 min. After centrifugation at 12,000× $g$ at 4 °C for 30 min, the concentration of protein supernatant was determined through the Bicinchoninic acid (BCA) method using the BCA Protein Assay Kit (Pierce, Thermo, USA). Protein samples at 100 μg, TEAB (Triethylammonium bicarbonate buffer), and TCEP (tris (2-carboxyethyl) phosphine) was combined for reaction for 60 min at 37 °C. IAM (Iodoacetamide) was added to the final concentration at 40 mM, and reacted for 40 min at room temperature under dark conditions. A certain percentage (acetone: sample v/v = 6:1) of pre-cooled acetone was added to each sample, then allowed to settle for 4 h at -20 °C. After centrifugation for 20 min at 10,000× $g$, the sediment was collected, and 100 μL of 100 mM TEAB solution was added. Finally, the mixture was digested with Trypsin overnight at 37 °C, and added at a 1:50 trypsin-to-protein mass ratio. Trypsin-digested peptides were analysed using online nano flow liquid chromatography tandem mass spectrometry performed on an EASY-nLC 1200 system (Thermo, USA) connected to a Q Exactive HF-X quadrupole orbitrap mass spectrometer (Thermo, USA) through a nanoelectrospray ion source. The data was uploaded to the Majorbio cloud platform (https://cloud.majorbio.com) after the database search for data analysis.

## Protein expression and purification

Plasmids pET28a-AcrA and pET28a-AcrB were constructed after codon optimization in *E. coli*. The plasmid was amplified in DH5α-competent cells, then transformed into BL21 (DE3)- competent cells. After overnight incubation, single colonies were picked and transferred to LB medium. The cultures were shaken at 37 °C for 4–6 h until the OD600 reached approximately 0.8, before induction of fusion protein expression by 0.1 mM isopropyl thio-β-d-galactoside (IPTG) at 37 °C for 5 h. Then, the cells were harvested by centrifugation and purified. Briefly, the cells were sonicated in PBS, the homogenate was centrifuged (12,000× $g$, 10 min, 4 °C), and the pellet was collected. The supernatant was passed through the Ni-NTA column at a flow rate of 1.0 mL/min by a peristaltic pump at 4 °C overnight, then passed onto the binding buffer (50 mM sodium phosphate buffer pH 8, 500 mM NaCl and 10 mM imidazole). The bacterial proteins were then removed with elution buffer A, which contained 30 mM imidazole (50 mM sodium phosphate buffer pH 8, 500 mM NaCl, and 30 mM imidazole) in 20 mL. Next, the column was eluted with elution buffer B and 200 mM imidazole (50 mM sodium phosphate buffer pH 8, 500 mM NaCl, and 200 mM imidazole) for 20 mL to obtain the target protein. The crude product was washed with PBS via a 10 kDa millipore centrifugal ultrafiltration tube to remove

imidazole. The fractions of pure product, total protein, supernatant, and precipitate were then analysed by SDS-PAGE and WB using an anti-His tag antibody (AE003, ABclonal, China).

## Real-time quantitative polymerase chain reaction (RT-qPCR)

Cellular total RNA was extracted using the RNA Isolater Total RNA Extraction Reagent (Vazyme, Cat. No. RC112-01), following the manufacturer's instructions. Complementary DNA (cDNA) was synthesized using HiScript II Q Select RT SuperMix for qPCR with gDNA wiper (Vazyme, Cat. No. R233-01) in a total volume of 10 μL. RT-qPCR was performed using 2× Universal SYBR Green Rapid qPCR Mix (Abclonal, Cat. No. RK21203) with 100 ng of cDNA and 5 nM primer pairs at a time. The results were monitored using the CFX96 Real-Time PCR Assay System (Bio-Rad, USA). Supplementary Table 3 lists all primers used in the quantitative PCR.

## Isothermal titration calorimetry (ITC)

To assess the interaction between the ligand and the protein, ITC experiments were conducted using MicroCal ITC at 25 °C. All titrations were performed at 25 °C while stirring at 300 rpm in PBS. A control experiment of titrant into buffer was performed to account for the heat of dilution. All titrations were repeated at least three times with similar results. For ligand-protein titrations, an approximate protein concentration of 50 μM was used. The concentration of the ligand is approximately ten times higher than that of the protein. All ITC experiments were carried out and analysed using Launch NanoAnalyze Software.

## Differential scanning fluorimetry (DSF)

For the DSF experiments, 20 μL samples were prepared in duplicate using 100 μM of protein and a compound concentration of 50 μM. The samples were heated from 20 to 95 °C with increments of 1 °C/min before incubation for 20 min, and fluorescence was measured at each step in nanoDSF (Nano Temper Prometheus NT.48, Germany). The change in melting temperature ($T_m$) values was calculated and recorded by the instrument. Data analysis and image generation were performed using PR. ChemControl Software.

## Haemolytic activity

Sterile defibrinated sheep haemocytes were washed three times with PBS, then 100 μL of 8% haemocytes was added to 100 μL of FLO or F8 at different concentrations (0, 1, 2, 4, 8, 16, 32, 64, 128, 256, and 512 μM). Meanwhile, 0.2% Triton X-100 and PBS were used as positive and negative controls, respectively. The solution was placed in a 96-well microtiter plate and was incubated at 37 °C for 1 h before centrifugation at 3000× $g$ for 10 min. Then, 100 μL of the supernatant was taken, and its absorbance was determined at 576 nm by an Infinity M200 Microplate reader (Tecan).

## Cytotoxicity

The cytotoxicity of FLO or F8 on pk15, Vero, and L-02 cells was estimated using the CCK-8 Cell Proliferation and Cytotoxicity Assay Kit (CCK-8, Solarbio, Cat. No. CA1210). Cells were seeded, counted, and evenly distributed into 96-well microtiter plates at a cell density of $10^5$ cells per well, and incubated for 1 day at 37 °C and 5% $CO_2$ to promote cell growth. Subsequently, cells were treated with different concentrations of FLO or F8, with six replicates for each condition. After incubating at 37 °C and 5% $CO_2$ for 24 h, 10 μL of CCK-8 solution was added to each well, then further incubated for 2 h. The absorbance was then measured at 450 nm using an Infinite M200 Microplate reader (Tecan).

## Resistance-development studies

A single colony of *E. coli* ATCC 25922 or *S. aureus* ATCC 29213 was picked and inoculated into MHB for overnight cultivation. Overnight

cultures were inoculated in fresh MHB containing 1/2, 1, 2, and 4×MIC of FLO or F8. The bacterial cultures were then incubated at 37 °C for 24 h at 200 rpm with continuous shaking. Subsequently, the MIC of bacteria from the second-highest concentrations with visible growth ($OD_{600}$ nm ≥0.3) was determined by broth microdilution in fresh MHB media containing different concentrations of FLO or F8. The cultures were serially passaged for 30 days.

## Animal studies

All experimental procedures were conducted in accordance with animal welfare guidelines, and were previously approved by the Animal Welfare and Ethics Committee of the Huazhong Agricultural University Wuhan, China (approval permit numbers: 202311010007 and 202311010008). All specific pathogen-free (SPF) Kunming mice (6–7 weeks old, weighing approximately 25 g) and SPF C57BL/6 mice (6–7 weeks old, weighing 18 ± 2 g) were purchased from Hubei Provincial Laboratory Animal Center (Wuhan, China). Animals were housed under standard humidity (50 ± 10%), temperature (25 ± 2 °C), and light-dark cycle (12 h each) conditions with free access to food and water.

To study the toxicity of FLO and F8, SPF Kunming mice were randomly divided into three groups and weighed once a day. Each group was given a dose of 1500 mg/kg of body weight (bw) of FLO or F8 for seven consecutive days, and their health conditions were observed daily. The control mice were given the same volume of solvent. These doses did not cause any deaths and were considered safe in this experiment. The mice were euthanized and dissected seven days later, and the blood, spleen, kidneys, thymus, liver, small intestine, and femur were removed and collected for further analysis. For acute toxicity assays of FLO and F8, survival and mortality status were recorded after a single administration of 5000 mg/kg bw, followed by 14 days of observation of the surviving mice once a day.

## Mouse systemic infection study

To evaluate the anti-infective effects of FLO and F8, a mouse systemic infection model was established. Briefly, the experiments were performed using SPF C57BL/6 mice (6–7 weeks old, weighing 18 ± 2 g), and the mice were infected by intraperitoneal injection with *S. aureus* ATCC 29213 at a dose of $5 \times 10^6$ CFU, or a pre-induced FLO-resistant *S. aureus* suspension (*n* = 6). One hour after the infection, mice were administered a dose of 60 mg/kg bw of either FLO or F8. The control mice were given an equivalent volume of the solvent. The health status of the mice was observed, and once an infected mouse died, the blood, liver, spleen, and kidneys were collected for subsequent analysis. At 24 h post-infection, surviving mice were euthanized by cervical dislocation, and the blood, liver, spleen, and kidneys were collected for bacterial CFU analysis and histopathological evaluation. Furthermore, the same mouse mice were given a single dose of FLO or F8 at a dose of 60 mg/kg bw, and the survival status of the mice was observed within 72 h (*n* = 10). The control mice were given the same volume of solvent.

## Histopathology

After decapitation, the tissues were immediately fixed in 4% paraformaldehyde in phosphate buffer and stored at 4 °C. For paraffin embedding, the organs were washed and dehydrated through a series of graded ethanol baths, followed by embedding in paraffin wax. Serial sections of 5 μm thickness were cut using a microtome and stained with haematoxylin-eosin (H&E). Sections were observed using a light microscope.

## TUNEL reaction

Serial sections of 5 μm thickness were cut using a microtome, and sections were deparaffinized and rehydrated. Proteins were digested by placing tissue sections in 20 mg/mL of proteinase K (Merck, Germany) and incubated at 37 °C for 15 min. Endogenous peroxidase was inactivated with 3% $H_2O_2$ in methanol for 10 min at room temperature, then the sections were incubated with 50 μL of the TUNEL reaction mixture at 37 °C for 1 h. They were protected from light, rinsed three times with PBS, and visualized under a fluorescence microscope with excitation wavelengths in the range of 450–500 nm and detection wavelengths in the range of 515–565 nm (green).

## Enzyme-linked immunosorbent assay (ELISA)

The tissues were weighed and homogenized in PBS. After centrifugation at 3000 rpm for 30 min, the supernatant was collected to quantify IL-6, IL-2, and Hsp70 levels with ELISA kits (MSKBIO, China), used according to the instructions.

## Statistical analyses

Each reaction was performed in triplicate, and the results are expressed as the mean ± standard deviation (SD). $P < 0.05$ was considered statistically significant (not significant [n.s], $0.01 \leq P < 0.05$ [*], $P < 0.01$ [**]). Statistical analyses and graphical presentations were performed using GraphPad Prism 8 (GraphPad Prism Inc., San Diego, CA, USA).

## Reporting summary

Further information on research design is available in the Nature Portfolio Reporting Summary linked to this article.

## Data availability

The data that support the findings of this study are available from the corresponding author on request. The transcriptomics data generated in this study have been deposited in the NCBI Sequence Read Archive (SRA) database under accession code PRJNA1127350 [https://www.ncbi.nlm.nih.gov/sra]. The metabolomics data generated in this study have been deposited in the MetaboLights database under accession code MTBLS10516 [https://www.ebi.ac.uk/metabolights/]. The mass spectrometry proteomics data in this study have been deposited to the ProteomeXchange Consortium (https://proteomecentral.proteomexchange.org) via the iProX partner repository with the dataset identifier PXD053342. Source data are provided with this paper.

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

## Acknowledgements

This work was supported by the National Key Research and Development Program of China (2018YFC1603005), National Natural Science Foundation of China (NSFC) (32072925), and Fundamental Research Funds for the Central Universities (2662020DKPY020).

## Author contributions

Jin Feng performed the major experiments and prepared the figures and tables. Youle Zheng participated in the synthesis of small molecule compounds. Youle Zheng and Wanqing Ma participated in the protein expression and purification experiments. Dapeng Peng, Yindi Xu, Defeng Weng and Zhifang Wang provided reagents. Xu Wang supervised the research, coordination and strategy. All authors provided critical revisions and approved the final manuscript.

## Competing interests

The authors declare no competing interests.
