## [Peer Review File · Nature Communications]

REVIEWER COMMENTS

Reviewer #1 (Remarks to the Author):

Wang and coworkers describe the fragment-based modification of the fenicol scaffold with the goal of improving activity against resistant bacteria. They arrive at compound F8, which has moderately improved potency (2-4x) compared to florfenicol against selected bacterial strains, including florfenicol-resistant strains. Multi-omics analysis indicates that several cellular pathways are affected by F8, including cell wall biosynthesis and amino acid biosynthesis. The authors hypothesize that, compared to other fenicols, F8 has gained a second mode of action by targeting *arcB*, an ornithine carbamoyl transferase. They conduct DSF and ITC experiments with the goal of supporting this hypothesis.

One of the main goals of this work is to design a hybrid antibiotic. However, the workflow the authors employ to reach F8 is to add fragments to the fenicol scaffold, not “hybridize” it to another antibiotic. The targeting of *arcB*, if it is a meaningful second target, seems to have occurred serendipitously, not by design. The design aspects were aimed at avoiding clashes within the bacterial ribosome. I think the narrative about hybrid antibiotics is not accurate and should be removed. Instead, this is a modified antibiotic that may have picked up a second cellular target.

The potency of F8 compared to florfenicol against most bacterial strains tested is often within the margin of error for MIC experiments (2x, or 1 two-fold dilution) (Table 1). The activity of florfenicol in the florfenicol-resistant strains (Table 3) was not given, and should be. Claims that this is a substantially more potent antibiotic than florfenicol should be removed.

It is rare for MBCs to be very close to MICs, yet almost all of the values in table 3 closely match the values in table 2. For phenicol derivatives, which are generally bacteriostatic antibiotics, MBCs are usually at least 2-4 fold MICs. Details should be given on how the MBC measurements were conducted, and a comment should be added as to why the MBCs are so low, even for the florfenicol control.

The DSF data does not provide convincing evidence that F8 binds to *arcB*. The thermal shift appears to be <1 °C. The ITC data does fit the correct pattern for binding to *arcB*.

Overall, it is not clear from the data provided that *arcB* is a substantial contributing factor to the slightly improved activity of F8 compared to florfenicol. Improved cellular accumulation, slightly improved binding to the ribosome, selective stalling of different nascent peptides, and several other factors seem just as likely as the acquisition of a second target. Inhibition of protein synthesis can explain much of the omics data and the ROS (see <https://doi.org/10.1016/j.jbc.2023.105163>). The main pieces of evidence that *arcB* is a significant target are computational docking and ITC/DSF.

An experiment that would help differentiate the contributions of the two mechanisms (*arcB* vs ribosome inhibition) would be to run MIC experiments in one (or, better, several) strains of bacteria that have constitutive *cfr* resistance. This should prevent both florfenicol and F8 from binding to the ribosome due to additional methylation of A2503. If *arcB* is a second target that contributes to the antibacterial activity of F8, then F8 should still show significant activity against these *cfr* strains, but florfenicol will show

greatly reduced activity. Alternatively, if *arcB* is not an essential gene under standard rich media growth conditions, then this experiment could be run under oxidative stress so reveal a difference in the two antibiotics (see <https://bmcmicrobiol.biomedcentral.com/articles/10.1186/1471-2180-9-183>).

A complimentary experiment would be to measure MICs in the *arcB* deletion mutant. These strains have been studied for over a decade (see previous link), and should be easy to generate or obtain. If *arcB* is a meaningful target under standard growth conditions, and is a second target for F8, then a difference in susceptibility might be seen between the F8 and florfenicol.

The mouse infection models that show a significant difference in activity of F8 vs florfenicol against florfenicol-resistant strains are compelling. It is possible that this is due to different PK properties of the two drugs, and not due to activity against the causative organisms of the infection. The *in vitro* susceptibilities of F8 and florfenicol should be provided for each of these strains (and for all of the strains in Table 3, as stated above). Additionally, PK properties of F8 and florfenicol should be measured side by side, which would add to the quality of the manuscript by showing clinical relevance.

The discovery of antibiotics that overcome resistance is a necessary pursuit. Some of the evidence in this manuscript indicate that F8 could be a first step towards a next-generation phenicol antibiotic. Since fenicols have fallen out of use in most countries due to toxicity concerns, this would have clinical impact. Since the primary short-term toxic effects of phenicols are in the liver and kidney, this should be investigated for F8. Further studies are needed to confirm that *arcB* is a significant secondary target of F8. Additionally, more explicit control experiments are required to determine the causes of the increased activity of F8 *in vivo* vs *in vitro*. Finally, the narrative of the paper that centers on hybrid antibiotics does not align well with the workflow of the paper, and should be substantially modified.

Reviewer #2 (Remarks to the Author):

Antimicrobial resistance (AMR) is a global threat to public health.

The authors describe a new synthetic antibiotic class that is potentially effective against a number of clinically relevant bacteria.

General Comments.

Results Section could be more effective if shortened to the most salient 5 to 6 Figs, focused on the design/synthesis; broad-spectrum antibacterial activity; minimal bacterial resistance; *in vivo* data; toxicity; mechanism of action

Results jargon needs to be defined for a general audience.

Discussion Section: repetitive, needs to focus on results in context of prior work

Specific Comments.

1. line 28, Abstract: ..*in vitro* and *in vivo* broad-spectrum antibacterial activity against..
2. line 31, Abstract: In a mouse model of drug-resistant bacteremia...
3. lines 73-85: Move *arcB* para to Discussion
4. line 74: ...biosynthesis and catabolism.

5. line 110: ...and in vivo antibacterial activity...
6. line 114: Gram-positive and Gram-negative throughout.
7. line 142: define atomic property field
8. line 153: define crash value
9. Tables 1-3: add legends
10. line 193: F8 MIC of 128 uM for Pseudomonas is 2-fold below that shown for others.
11. Label figures 1-10.
12. Fig. 2 can be condensed to show principal in vivo data: CFU, and tissue data (Tables 1-3 in vitro data).
13. Fig. 1c is a table that should be moved to supplementary material
14. Fig. 3bc: define spleen index and thymus index
15. Fig. 3gh: for solid tissues, pg/mL should be expressed as pg/g
16. Fig. 4i: state route used for drug delivery
17. Fig. 5-8 can be condensed/summarized or placed in Suppl.
18. line 237: IL-2 and Hsp70 (Fig. 3ef) both show significant difference between PBS and F8 samples
19. line 245: HGB (Fig3o) shows significant difference between PBS and F8 samples
20. line 249-253: toxicity data in supplemental Fig3a, Fig4a-c, Fig5a belongs in main figs
21. lines 319-321: repetitive from 317-319.
22. lines 327: Fig. 9c is referred to before (Fig. 9a,b);
23. label Fig. 9c w/gene expression; define groups 1-3 in legend
24. Fig. 9d: state composition of the negative control; indicate expected size of ArcB
25. Fig. 9g typo: blank-arcB
26. lines 364-366: define permeability by PI staining; Fig. 10c,d; label with PI staining;
27. Fig. 10ef: Explain pH measurement; label w/pH
28. Fig. 10gh: Explain ROS measurement; label w/ROS
29. Fig. 10ij: Explain ATP measurement

Reviewer #3 (Remarks to the Author):

Noteworthy Results: The manuscript presents the development of a new antibiotic, F8, effective against various antibiotic-resistant bacteria. The identification of a novel target, ornithine carbamoyl transferase (arcB), and the use of multi-omics analysis are key highlights.

Significance to the Field: The work is significant due to its potential impact on addressing antibiotic resistance, a major global health issue. The approach to drug design and target identification could influence future research in the field.

Comparison to Established Literature: The manuscript appears original in its approach to combining structural hybridization-based drug design with multi-omics analysis for target identification. This approach is distinct from traditional methods in antibiotic research.

Support for Conclusions: The conclusions are supported by comprehensive data, including in vitro and in vivo studies, molecular docking, and biochemical assays. However, further research and clinical trials might be needed to fully validate the drug's effectiveness and safety.

Flaws in Data Analysis or Interpretation: The analysis and interpretation of data appear robust. However, as with any scientific study, independent replication and additional studies could further validate the findings.

Soundness of Methodology: The methodology, encompassing drug design, target identification, and validation, are sound and adheres to current standards in the field.

Detail for Reproduction: The manuscript provides sufficient detail for the methods, allowing for potential reproduction of the research, which is crucial for scientific validation.

Minor comments:

Expanded Discussion on Limitations: More detailed discussion on potential limitations of F8, including any observed side effects, resistance development potential, and effectiveness in diverse biological environments.

Comparison with Existing Antibiotics: Provide a more comprehensive comparison of F8 with existing antibiotics, particularly those targeting similar bacteria, to contextualize its efficacy and potential advantages.

Environmental Impact Assessment: Discuss any potential environmental impacts of F8, particularly if used widely, considering the ongoing concerns about antibiotic pollution.

Mechanism of Resistance Development: Could there be a discussion on how bacteria might develop resistance to F8, as understanding resistance mechanisms is key to developing more effective antibiotics.

Title: A synthetic antibiotic class with a deeply-optimized design for overcoming bacterial resistance

The following is a point-to-point response to the Reviewers' comments.

Reviewer #1 (Remarks to the Author):

Wang and coworkers describe the fragment-based modification of the fenicol scaffold with the goal of improving activity against resistant bacteria. They arrive at compound F8, which has moderately improved potency (2-4×) compared to florfenicol against selected bacterial strains, including florfenicol-resistant strains. Multi-omics analysis indicates that several cellular pathways are affected by F8, including cell wall biosynthesis and amino acid biosynthesis. The authors hypothesize that, compared to other fenicols, F8 has gained a second mode of action by targeting arcB, an ornithine carbamoyl transferase. They conduct DSF and ITC experiments with the goal of supporting this hypothesis.

The Authors' Response: We highly appreciate your time and efforts in reviewing our manuscript and your insightful comments and suggestions for our work. We provided new data and information below and in the revised manuscript to address all your concerns about the important technical points.

1. **The Reviewer's Comment:** One of the main goals of this work is to design a hybrid antibiotic. However, the workflow the authors employ to reach F8 is to add fragments to the fenicol scaffold, not "hybridize" it to another antibiotic. The targeting of arcB, if it is a

meaningful second target, seems to have occurred serendipitously, not by design. The design aspects were aimed at avoiding clashes within the bacterial ribosome. I think the narrative about hybrid antibiotics is not accurate and should be removed. Instead, this is a modified antibiotic that may have picked up a second cellular target.

The Authors' Response: We are extremely grateful for your constructive comments and advice. We sincerely apologize for our oversight on this matter. With regard to your reference to adding fragments to the florfenicol scaffold rather than hybridizing with another antibiotic, we recognize that this may have led to a misunderstanding of the definition of "hybrid antibiotics". Indeed, our approach is more accurately described as a modification of the existing antibiotic florfenicol by adding specific fragments to enhance its efficacy and extend its mechanism of action, rather than by directly combining the properties of two different antibiotics. We fully agree with the Reviewer's suggestion that it would be more accurate to revise our description to "modified antibiotics" to avoid misuse of the concept of "hybrid antibiotics". This indicates that the antimicrobial mechanism of F8 does not depend entirely on ribosome inhibition and that *arcB* is an important target in the antimicrobial activity of F8.

In the revised manuscript, we clearly state that our approach is a structural modification of the florfenicol antibiotic, rather than the creation of a "hybrid antibiotics" in the traditional sense. We have amended the sentence from "hybrid structures" to "**structural modification**". We have removed the description of hybrid antibiotics and adjusted the use of related terminology to ensure that our descriptions are more accurate and clear.

Please kindly review the changes in the revised manuscript. The quality of our manuscript has greatly improved thanks to your insights. We are profoundly grateful.

2. The Reviewer's Comment: The potency of F8 compared to florfenicol against most bacterial strains tested is often within the margin of error for MIC experiments (2×, or 1 two-fold dilution) (Table 1). The activity of florfenicol in the florfenicol-resistant strains (Table 3) was not given, and should be. Claims that this is a substantially more potent antibiotic than florfenicol should be removed.

The Authors' Response: We are extremely grateful for your valuable suggestions and sincerely apologize for the oversight regarding the lack of data on the activity of florfenicol in resistant strains (Table 3), a mistake that indeed reduced the transparency and verifiability of our study. Thank you very much for pointing it out. We have reviewed and validated our experimental data, supplementing this information in Table 3 of the revised manuscript. Ensuring the completeness and transparency of our research results is our top priority, and we appreciate your help in this regard.

We agree with your suggestion to remove the description that F8 is a substantially more potent antibiotic than florfenicol. Prompted by your insights, we have carefully revised our manuscript to express our findings more cautiously, recognizing that our preliminary description may have been too absolute without the support of more extensive data.

Please kindly review the changes in the revised manuscript. The quality of our manuscript has greatly improved thanks to your insights. We are profoundly grateful.

Table 3. MIC of F8 against antibiotic-resistant strains.

Organism	MIC (μM)	
	F8	FLO
Methicillin-resistant S. aureus B1-1	8	32
Polymyxin B-resistant E. hormaechei wb 4	16	128
Florfenicol-resistant S. suis 1136	8	64

Florfenicol-resistant S. suis 1194	4	16
Florfenicol-resistant S. suis 1197	8	32
Florfenicol-resistant S. suis 1655	8	32
Florfenicol-resistant S. suis 1658	4	32
Florfenicol-resistant S. suis 1669	16	64
Florfenicol-resistant H. parasuis 1565	8	32
Florfenicol-resistant H. parasuis 1614	4	16
Florfenicol-resistant H. parasuis 1651	8	32
Ampicillin-resistant S. typhi BYG 9	4	64
Sulfamethoxazole-resistant S. typhi BYG 21	4	32
Doxycycline-resistant S. typhi BYG 11	4	32
Doxycycline-resistant S. typhi BYG 25	8	32
Doxycycline-resistant S. typhi BYG 31	8	32

The MICs of antibiotics were determined by the broth microdilution method (see materials and methods) following the Clinical and Laboratory Standards Institute (CLSI) guidelines. The assay was done three times to confirm results (n=3). Source data are provided as a Source Data file. MIC, Minimum Inhibitory Concentration; FLO, florfenicol.

3. The Reviewer's Comment: It is rare for MBCs to be very close to MICs, yet almost all of the values in table 3 closely match the values in table 2. For phenicol derivatives, which are generally bacteriostatic antibiotics, MBCs are usually at least 2-4 fold MICs. Details should be given on how the MBC measurements were conducted, and a comment should be added as to why the MBCs are so low, even for the florfenicol control.

The Authors' Response: We greatly appreciate your comprehensive and helpful critique. We sincerely apologize for our oversight on this matter. In response to the Reviewer's questions about the phenomenon of MBC values close to MIC values and the experimental methodology, we provide the following detailed responses and additions.

1) Firstly, we reconfirmed the determination method for MBC as follows: After determining the minimum inhibitory concentration (MIC), we took 10 μ L of bacterial suspension from each antibiotic concentration that did not show growth, and inoculated it onto fresh MH agar plates without antibiotics for further cultivation. Bacterial growth was

then assessed after 24 hours of incubation. The MBC value for each antibiotic was determined at the lowest antibiotic concentration at which no bacterial growth was observed. It should be noted that we followed the Clinical and Laboratory Standards Institute (CLSI) guidelines to ensure standardization and comparability of our methods.

In the revised manuscript, we have inserted the following content in **Methods**, "**The MBC values were identified as the lowest drug concentration to kill over 99.9% of bacteria. An aliquot of 10 μ L bacterial suspension from each well was plated onto MH agar plates. After the plates were incubated at 37°C for 24 h, MBC values were determined by visual inspection of CFU on the agar**".

2) Regarding the phenomenon of MBC values close to MIC values, we further considered and analyzed the following possible explanatory factors:

Incubation conditions: Unsuitable incubation times may have affected the growth of the bacteria, resulting in inaccuracies in the reading of MIC and MBC values. In our previous assay, in which an equal amount of 10 μ L of bacterial suspension was taken from each well and inoculated it onto MH agar plates, was incubated at 37°C for 8 h. The incubation time did not reach 24 h, which may have resulted in errors in the MBC values.

Preparation of bacterial suspension: Incorrect or inconsistent densities of the inoculated bacterial suspension could influence the test outcomes. In our previous detection method, the detection of MIC and MBC values for some bacteria was completed in two stages, before and after. Variability in the growth status and density of different batches of bacterial suspensions may result in inaccuracies in MBC values. We sincerely apologize for the mistakes made in the experimental operation.

For control florfenicol, we also observed that the MBC values were close to the MIC values. We referred to the relevant literature and compared it with the available data. Several studies have shown that the MIC and MBC values for florfenicol against *P. multocida* are very similar, with MBC:MIC ratios less than 2:1. Although uncommon, this phenomenon may indeed occur under certain conditions.

3) To ensure the accuracy and repeatability of the results, we conducted multiple verifications of key steps in the experimental process, including the preparation of antibiotic solutions, standardization of bacterial inoculation, and consistency of incubation conditions. In addition, we increased the number of experimental repetitions and had an independent team reproduce some of the experimental results to ensure their reliability. We verified and modified the MBC values of the compounds in the results section. We hope that these additional details and analyses will satisfy the Reviewer's requirements for experimental methods and interpretation of results.

Please kindly review the changes in the revised manuscript. The quality of our manuscript has greatly improved thanks to your insights. We are profoundly grateful.

Table 2. MBC of broad-spectrum antibacterial small molecule compounds based on structural hybridization strategy.

Organism	MBC (μM)											
	FLO	F1	F2	F3	F4	F6	F7	F8	F11	F13	F14	F15
E. coli	16	32	64	>128	>128	16	32	8	>128	>128	32	>128
S. typhi	8	16	64	>128	>128	8	32	4	>128	>128	16	>128
P. aeruginosa	>128	>128	>128	>128	>128	>128	>128	>128	>128	>128	>128	>128
S. aureus	32	64	64	>128	>128	32	64	16	>128	>128	64	>128
B. subtilis	8	16	32	>128	>128	8	8	4	>128	>128	16	>128

E. faecalis	32	64	64	>128	>128	16	16	8	>128	>128	32	>128
P. multocida	8	16	8	>128	128	4	8	4	>128	>128	4	>128
A. pleuropneumoniae	32	64	64	>128	>128	8	64	8	>128	>128	32	>128
S. suis	32	128	128	>128	>128	16	32	8	>128	>128	64	>128
H. parasuis	64	128	>128	>128	>128	32	64	16	>128	>128	64	>128

The MBCs of antibiotics were determined as the lowest concentration of tested compound that killed at least 99.9% of the initial inoculums. The assay was done three times to confirm results (n=3). MBC, Minimum Bactericidal Concentration; FLO, florfenicol.

4. The Reviewer's Comment: The DSF data does not provide convincing evidence that F8 binds to arcB. The thermal shift appears to be <1°C. The ITC data does fit the correct pattern for binding to arcB.

The Authors' Response: We are extremely grateful for your constructive comments and advice. We appreciate your pointing out the potential limitations of Differential Scanning Fluorimetry (DSF) data in demonstrating the binding of F8 to arcB and understand your concern regarding the relatively small thermal shift (<1°C).

To more comprehensively elucidate the DSF data, we have conducted additional analyses and extensively reviewed the literature on interactions between molecules of similar size and properties. In these instances, thermal shifts even smaller than 1°C are considered meaningful evidence of binding. It has been accepted that ΔT_m values correlated with hit-likeness shown below¹:

For more ligandable binding sites, the grouping of ΔT_m values might be as follows: >2.5°C—strong hit; 1–2.5°C—medium hit; 0.5–1°C—weak hit.

For less ligandable binding sites, the grouping of ΔT_m values might be as

follows: $>1.5^{\circ}\text{C}$ —strong hit; $0.5\text{--}1^{\circ}\text{C}$ —medium hit; $0.2\text{--}0.5^{\circ}\text{C}$ —weak hit.

It has been accepted that fragments were considered to be hits when ΔT_m was higher than 0.5°C ^{2, 3}. Amaning et al. showed⁴ that both ligands, fragments 18 and 19, had ΔT_m values of 0.8°C and were able to bind to the kinase MEK1. Schuller et al. showed⁵ that both ligands, ZINC331945 and ZINC26180281, had ΔT_m values of 0.6°C and were able to bind to Mac1, a conserved large structural domain of coronaviruses. Furthermore, Looock et al. found⁶ that the incubation of M protein with JNJ-9676 yielded a 0.9°C stabilization in ΔT_m , confirming the drug-target interaction.

In our experiments, we observed a thermal shift of 0.8°C in the DSF assay. Based on the literature, thermal shifts greater than 0.5°C generally indicate the presence of weak binding. We have further corroborated this conclusion with isothermal titration calorimetry (ITC) data, which confirmed the binding of F8 to the arcB protein.

We greatly appreciate your valuable suggestions. We believe that an in-depth discussion of these issues will not only strengthen the conclusions of the current study, but also guide our future work. We look forward to receiving your further guidance and suggestions.

5. The Reviewer's Comment: Overall, it is not clear from the data provided that arcB is a substantial contributing factor to the slightly improved activity of F8 compared to florfenicol. Improved cellular accumulation, slightly improved binding to the ribosome, selective stalling of different nascent peptides, and several other factors seem just as likely as the acquisition of a second target. Inhibition of protein synthesis can explain much of the omics data and the ROS (see <https://doi.org/10.1016/j.jbc.2023.105163>). The main pieces of evidence that arcB is

a significant target are computational docking and ITC/DSF.

The Authors' Response: We are extremely grateful for your constructive comments and advice. We have addressed the Reviewer's inquiries and have made revisions and additions to this section accordingly. Looking at the available data, both computational docking and ITC/DSF experiments suggest a specific interaction between arcB and F8. This interaction suggests that F8 may exert part of its antimicrobial effect by binding to arcB.

We agree with the Reviewer's comments and have added additional experiments based on their suggestions to further demonstrate that arcB is a crucial factor in the increased F8 activity.

In accordance with the Reviewer's suggestions, we selected three bacterial strains with constitutive cfr resistance for our MIC experiments. The cfr gene confers resistance to certain antibiotics through specific rRNA methylation at the A2503 site. We observed that F8 still maintained significant activity in these strains with cfr resistance, with MIC values of 16-32 μM , while the activity of florfenicol was significantly reduced, with MIC values $>128 \mu\text{M}$, suggesting that arcB is a crucial target in the antimicrobial activity of F8. The activity of florfenicol was significantly reduced in these strains, which further confirms that its action is mainly dependent on interactions with the ribosome, being affected by cfr-induced A2503 methylation. These findings underscore that the antimicrobial mechanism of F8 is not solely reliant on ribosome inhibition, offering new insights into the significant role of arcB in the antimicrobial activity of F8.

Furthermore, we examined the antimicrobial activity of F8 against the arcB deletion mutant. We obtained the ΔarcB mutant *P. multocida* from the Ma research group at Huazhong

Agricultural University and tested the MIC values. Our results show that F8 has an attenuated antimicrobial effect in the *arcB* deletion mutant with a MIC of 16 μM , whereas florfenicol has an unchanged MIC of 8 μM , further suggesting that *arcB* is an effective target for F8. At the same time, we did not observe significant changes in susceptibility to florfenicol in the *arcB* deletion mutant. This could also explain the potential difference in antimicrobial activity between F8 and florfenicol.

The detailed content and results of these experiments have been included in the revised manuscript. Your insightful suggestions have significantly contributed to our study, advancing our understanding of the antimicrobial mechanism of F8. The quality of our manuscript has been greatly improved. We are profoundly grateful.

Fig 5. Binding of F8 with arcB. **a**, Experimental schematic for the multi-omics experiment. **b**, Gene-level differences of arcB in *S. aureus* treated with F8 or positive control FLO. Groups 1-3 represent three sets of transcriptomic samples, respectively. **c**, Simulated image of the F8 4Å DBD in arcB (PDB:2ksd) generated by PyMOL software, with residues LEU-31 shown in green. **d**, The superposition of the crystal structure of the DBDs of F8 and FLO bound to arcB, with residues LEU-31 shown in green, FLO in light purple, and F8 in blue. **e**, The expression of arcB was analysed by SDS-PAGE, and compared with the negative control (C). The target protein was expressed in the form of total protein (T), soluble (S), and insoluble (P) of the lysate. **f, g**, ITC detection of ligand target-binding affinity. **h**, Detection of thermal displacement (T_m change) by DSF. **i**, MIC of F8 against strains with constitutive cfr resistance. **j**, MIC of F8 against the arcB deletion mutant. The assay was done three times to confirm results ($n = 3$). FLO, florfenicol.

6. The Reviewer's Comment: An experiment that would help differentiate the contributions of the two mechanisms (arcB vs ribosome inhibition) would be to run MIC experiments in

one (or, better, several) strains of bacteria that have constitutive cfr resistance. This should prevent both florfenicol and F8 from binding to the ribosome due to additional methylation of A2503. If arcB is a second target that contributes to the antibacterial activity of F8, then F8 should still show significant activity against these cfr strains, but florfenicol will show greatly reduced activity. Alternatively, if arcB is not an essential gene under standard rich media growth conditions, then this experiment could be run under oxidative stress so reveal a difference in the two antibiotics (see <https://bmcmicrobiol.biomedcentral.com/articles/10.1186/1471-2180-9-183>).

The Authors' Response: We are extremely grateful for your constructive comments and advice. We have addressed the Reviewer's inquiries and have made revisions and additions to this section. The suggestion to conduct MIC experiments in bacterial strains with constitutive cfr resistance, to distinguish the effects of arcB gene action from cfr-mediated ribosomal inhibition, is extremely valuable. It significantly enhances the depth and breadth of our research. We have carefully reviewed the literature you recommended and have enriched our research program.

In accordance with the Reviewer's suggestions, we selected three bacterial strains with constitutive cfr resistance for our MIC experiments. The cfr gene confers resistance to certain antibiotics through specific rRNA methylation at the A2503 site. We observed that F8 still maintained significant activity in these strains with cfr resistance, with MIC values of 16-32 μM , while the activity of florfenicol was significantly reduced, with MIC values $>128 \mu\text{M}$, suggesting that arcB is a crucial target in the antimicrobial activity of F8. The activity of florfenicol was significantly reduced in these strains, which further confirms that its action is

mainly dependent on interactions with the ribosome, being affected by *cfr*-induced A2503 methylation. These findings underscore that the antimicrobial mechanism of F8 is not solely reliant on ribosome inhibition, offering new insights into the significant role of *arcB* in the antimicrobial activity of F8.

In the revised manuscript, we have added this section of data in **Fig.5i** and inserted the following content in the **Results**, "**To further confirm that *arcB* is an important factor in the increase of F8 activity, we selected three bacterial strains with constitutive *cfr* resistance for MIC experiments. The results showed that F8 still maintained significant activity in these strains with *cfr* resistance, with MIC values of 16-32 μ M, while the activity of FLO was significantly reduced, with MIC values >128 μ M (Fig. 5i). This indicates that the antimicrobial mechanism of F8 does not depend entirely on ribosome inhibition and that *arcB* is an important target in the antimicrobial activity of F8.**" In addition, we have carefully adjusted and supplemented the content related to this section throughout the entire text.

Please kindly review the changes in the revised manuscript. The quality of our manuscript has greatly improved thanks to your insights. We are profoundly grateful.

Organism	F8	FLO
S. aureus 0705K2	32	>128
S. aureus 0705205	16	>128
S. aureus 0705220	32	>128

Fig. 5i, MIC of F8 against strains with constitutive *cfr* resistance. The assay was done three times to confirm results (n = 3). FLO, florfenicol.

7. The Reviewer's Comment: A complimentary experiment would be to measure MICs in the *arcB* deletion mutant. These strains have been studied for over a decade (see previous link), and should be easy to generate or obtain. If *arcB* is a meaningful target under standard growth conditions, and is a second target for F8, then a difference in susceptibility might be seen between the F8 and florfenicol.

The Authors' Response: We are extremely grateful for your constructive comments and advice. We sincerely apologize for our oversight on this matter. We agree with the Reviewer's suggestions to measure MICs in the *arcB* deletion mutant. Your advice is immensely helpful for us to gain a deeper understanding of the antimicrobial mechanisms of F8 against bacteria and to validate the effectiveness of *arcB* as a potential target of action.

In accordance with the Reviewer's suggestions, we obtained the Δ *arcB* mutant *P. multocida* from the Lv research group⁷ at Huazhong Agricultural University and tested the MIC values. Our results show that F8 has an attenuated antimicrobial effect in the *arcB* deletion mutant with a MIC of 16 μ M, whereas florfenicol has an unchanged MIC of 8 μ M, further suggesting that *arcB* is an effective target for F8. At the same time, we did not observe significant changes in susceptibility to florfenicol in the *arcB*-deficient strains. This could also explain the potential difference in antimicrobial activity between F8 and florfenicol.

In the revised manuscript, we have added this section of data in **Fig.5j** and inserted the following content in the **Results**, "**Furthermore, we examined the antimicrobial activity of F8 against the *arcB* deletion mutant. The results showed that F8 has an attenuated antimicrobial effect in the Δ *arcB* mutant *P. multocida* with a MIC of 16 μ M, whereas**

FLO has an unchanged MIC of 8 μ M (Fig. 5j), further suggesting that arcB is an effective target for F8." In addition, we have carefully adjusted and supplemented the content related to this section throughout the entire text.

Please kindly review the changes in the revised manuscript. The quality of our manuscript has greatly improved thanks to your insights. We are profoundly grateful.

Organism	F8	FLO
Δ arcB mutant P. multocida	16	8
P. multocida	4	8

Fig. 5j, MIC of F8 against the arcB deletion mutant. The assay was done three times to confirm results (n = 3). FLO, florfenicol.

8. The Reviewer's Comment: The mouse infection models that show a significant difference in activity of F8 vs florfenicol against florfenicol-resistant strains are compelling. It is possible that this is due to different PK properties of the two drugs, and not due to activity against the causative organisms of the infection. The in vitro susceptibilities of F8 and florfenicol should be provided for each of these strains (and for all of the strains in Table 3, as stated above). Additionally, PK properties of F8 and florfenicol should be measured side by side, which would add to the quality of the manuscript by showing clinical relevance.

The Authors' Response: We are extremely grateful for your constructive comments and advice. Your insights are critically important for the deepening and refinement of our research.

In accordance with the Reviewer's suggestions, we have reviewed and provided the *in vitro* drug sensitivity of each strain to F8 and florfenicol. The results showed that F8

significantly inhibited florfenicol-resistant *S. aureus* and florfenicol-resistant *E. coli* (**Fig. 4c**). There were also significant differences in the activity of F8 against florfenicol-resistant strains in the mouse infection model (**Fig. 4d–i**). These data have been provided in the revised manuscript. This will help us to better understand that the difference in the activity of the two drugs against resistant strains may be a difference in their own activity against the pathogenic bacteria, rather than a difference in PK characteristics. In addition, we supplemented the *in vitro* drug susceptibility testing of F8 and florfenicol against the strains mentioned in Table 3 to ensure the comprehensiveness and transparency of the study. These results facilitate a direct comparison of the activities of the two drugs against different strains, which have been added to **Table 3** in the revised manuscript. We believe that these *in vitro* data provide compelling preliminary evidence of the antimicrobial activity of F8.

We fully agree with the Reviewer's perspective on the importance of pharmacokinetic (PK) data in evaluating the antimicrobial activity of F8 and florfenicol. However, in designing this study, our initial intention was to concentrate on the drug's *in vitro* antibacterial efficacy, aimed at first identifying candidates with potential therapeutic value. To ensure clarity in research focus, we focused on validating the antibacterial activity of F8 *in vitro* and *in vivo* models. The comparison of the antimicrobial activity of the two drugs in both *in vitro* and *in vivo* settings suggests that the observed differences may be attributed to their intrinsic activity against the pathogens, rather than differences in their PK properties. In fact, PK experiments have already been conducted as part of a separate follow-up research project, which also includes pharmacodynamics (PD) and long-term safety assessments among other preclinical experiments. We are very eager to share our new findings with the academic community upon

completion.

We greatly appreciate your valuable suggestions. We believe that an in-depth discussion of these issues will not only strengthen the conclusions of the current study, but also guide our future work. We look forward to receiving your further guidance and suggestions.

Fig 4c. Inhibition of florfenicol-resistant *S. aureus* and florfenicol-resistant *E. coli* by different concentrations of F8.

Fig 4. F8 overcomes resistance and is effective *in vivo*. **d–g,** Bacterial load in the blood, liver, spleen, and kidneys of different treatment groups in a mouse model of *S. aureus* bacteraemia (n = 6). ns, not significant, *, $p < 0.05$, **, $p < 0.01$, ***, $p < 0.001$. **h,** Histopathological assessment of the liver, spleen, and kidneys in different treatment groups in a mouse model of florfenicol-resistant *S. aureus* bacteraemia. Scale bar, 100 μm . **i,** Survival curves of mice in an anti-*S. aureus* study (n = 10). F8/FLO were administered via oral gavage as the route of delivery.

Table 3. MIC of F8 against antibiotic-resistant strains.

Organism	MIC (μ M)	
	F8	FLO
Methicillin-resistant S. aureus B1-1	8	32
Polymyxin B-resistant E. hormaechei wb 4	16	128
Florfenicol-resistant S. suis 1136	8	64
Florfenicol-resistant S. suis 1194	4	16
Florfenicol-resistant S. suis 1197	8	32
Florfenicol-resistant S. suis 1655	8	32
Florfenicol-resistant S. suis 1658	4	32
Florfenicol-resistant S. suis 1669	16	64
Florfenicol-resistant H. parasuis 1565	8	32
Florfenicol-resistant H. parasuis 1614	4	16
Florfenicol-resistant H. parasuis 1651	8	32
Ampicillin-resistant S. typhi BYG 9	4	64
Sulfamethoxazole-resistant S. typhi BYG 21	4	32
Doxycycline-resistant S. typhi BYG 11	4	32
Doxycycline-resistant S. typhi BYG 25	8	32
Doxycycline-resistant S. typhi BYG 31	8	32

The MICs of antibiotics were determined by the broth microdilution method (see materials and methods) following the Clinical and Laboratory Standards Institute (CLSI) guidelines. The assay was done three times to confirm results (n=3). Source data are provided as a Source Data file. MIC, Minimum Inhibitory Concentration; FLO, florfenicol.

9. The Reviewer's Comment: The discovery of antibiotics that overcome resistance is a necessary pursuit. Some of the evidence in this manuscript indicate that F8 could be a first step towards a next-generation phenicol antibiotic. Since fenicols have fallen out of use in most countries due to toxicity concerns, this would have clinical impact. Since the primary short-term toxic effects of phenicols are in the liver and kidney, this should be investigated for F8. Further studies are needed to confirm that arcB is a significant secondary target of F8. Additionally, more explicit control experiments are required to determine the causes of the increased activity of F8 *in vivo* vs *in vitro*. Finally, the narrative of the paper that centers on hybrid antibiotics does not align well with the workflow of the paper, and should be

substantially modified.

The Authors' Response: We are extremely grateful for the Reviewer's attention and recognition of our manuscript, which is greatly encouraging to us. We fully agree with the importance of discovering antibiotics to overcome drug resistance. Our study provides some evidence that F8 may be the first step towards a next-generation phenicol antibiotic. Considering the clinical limitations of fenicol, we believe that the development of F8 is of great significance.

We agree that fenicol is no longer used in most countries due to toxicity concerns. This is also one of the primary motivations for our research on F8. We have studied the toxic effects of F8, including organ toxicity to the liver, kidneys, intestines, spleen, thymus, and bone marrow, while also examining the hematotoxicity and immunotoxicity of F8. Research has shown that there are no significant lesions in the liver and kidneys of mice treated with F8. F8 exhibited excellent tolerance in a 14-day study of acute toxicity in mice, with a maximum dose of 5000mg/kg. In addition, F8 has low cytotoxicity to mammalian cells and low hemolysis rate to sheep red blood cells. We have carefully adjusted and supplemented the content related to this section throughout the text (see **Fig. 3** for details).

We appreciate the Reviewer's suggestion that further studies are needed to confirm that *arcB* is a significant secondary target of F8. In accordance with the Reviewer's suggestions, we selected three bacterial strains with constitutive *cfr* resistance for our MIC experiments. Furthermore, we examined the antimicrobial activity of F8 against the *arcB* deletion mutant, further suggesting that *arcB* is an effective target for F8. We have carefully adjusted and supplemented the content related to this section throughout the text (see **Fig. 5i, j** for details).

We recognize the need for more explicit control experiments to determine the causes of the increased activity of F8 *in vivo* vs *in vitro*. In accordance with the Reviewer's suggestions, we have reviewed and provided the *in vitro* drug sensitivity of each strain to F8 and florfenicol. The results showed that F8 significantly inhibited florfenicol-resistant *S. aureus* and florfenicol-resistant *E. coli* (**Fig. 4c**). There were also significant differences in the activity of F8 against florfenicol-resistant strains in the mouse infection model. These data have been provided in the revised manuscript. This will help us better understand the differences in the activity of the two drugs against resistant strains. In addition, we supplemented the *in vitro* drug susceptibility testing of F8 and florfenicol against the strains mentioned in Table 3. These results facilitate a direct comparison of the activities of the two drugs against different strains, which have been added to **Table 3** in the revised manuscript. We believe that these *in vitro* data provide compelling preliminary evidence of the antimicrobial activity of F8.

We fully recognize the narrative of the paper that centers on hybrid antibiotics does not align well with the workflow of the paper. We thoroughly reviewed and adjusted the manuscript to accurately reflect the research methodology and findings. In the revised manuscript, we clearly state that our approach is a structural modification of the florfenicol antibiotic, rather than the creation of a "hybrid antibiotics" in the traditional sense. We have amended the sentence from "hybrid structures" to "**structural modification**". We have removed the description of hybrid antibiotics and adjusted the use of related terminology to ensure that our descriptions are more accurate and clear.

We sincerely appreciate your constructive feedback, which will undoubtedly enhance the

quality and impact of our manuscript. Your insights will provide us with valuable guidance as we endeavor to advance research in the field of antibiotic discovery.

Fig 3. Safety evaluation of F8. **a**, Spleen index. Spleen index (mg/g) = spleen weight (mg)/body weight of mice (g). **b**, Thymus index. Thymus index (mg/g) = thymus weight (mg)/body weight of mice (g). **c**, Concentration of IL-6 in serum. **d**, Concentration of IL-2 in serum. **e**, Concentration of Hsp70 in serum. **f-h**, Concentration of Hsp70 in the spleen, thymus, and bone marrow. ns, not significant, *, $p < 0.05$, **, $p < 0.01$, ***, $p < 0.001$. **i**, Histopathological assessment of the spleen, thymus, and bone marrow. Scale bar, 100 μm . **j**, Apoptotic cells in the spleen, thymus, and bone marrow in different groups (TUNEL staining, under a fluorescence microscope at 200 \times). **k**, Acute toxicity test, record of mouse body weight after a single dose of F8 or positive control FLO at 5000 mg/kg (n=6). **l-n**, Toxicity of different concentrations of PBS of F8 or positive control FLO (0 – 256 μM) on the Vero cells, L-02 cells and pk15 cells (n=6). **o**, Hemolysis rate of sheep red blood cells by different concentrations of F8 or positive control FLO (0 – 256 μM) (n=6).

References

- 1 Coyle, J. & Walser, R. Applied Biophysical Methods in Fragment-Based Drug Discovery. *SLAS discovery : advancing life sciences R & D* **25**, 471-490, doi:10.1177/2472555220916168 (2020).
- 2 Kitel, R. *et al.* Exploring the Surface of the Ectodomain of the PD-L1 Immune

- Checkpoint with Small-Molecule Fragments. *ACS chemical biology* **17**, 2655-2663, doi:10.1021/acscchembio.2c00583 (2022).
- 3 Silvestre, H. L., Blundell, T. L., Abell, C. & Ciulli, A. Integrated biophysical approach to fragment screening and validation for fragment-based lead discovery. *Proceedings of the National Academy of Sciences of the United States of America* **110**, 12984-12989, doi:10.1073/pnas.1304045110 (2013).
- 4 Amaning, K. *et al.* The use of virtual screening and differential scanning fluorimetry for the rapid identification of fragments active against MEK1. *Bioorganic & medicinal chemistry letters* **23**, 3620-3626, doi:10.1016/j.bmcl.2013.04.003 (2013).
- 5 Schuller, M. *et al.* Fragment binding to the Nsp3 macrodomain of SARS-CoV-2 identified through crystallographic screening and computational docking. *Science advances* **7**, doi:10.1126/sciadv.abf8711 (2021).
- 6 Loock, M. V. *et al.* A small-molecule sars-cov-2 inhibitor targeting the membrane protein. Preprint at <https://doi.org/10.21203/rs.3.rs-3975125/v1> (2024).
- 7 Lv, Q. *et al.* Identification of two-component system ArcAB and the universal stress protein E in *Pasteurella multocida* and their effects on bacterial fitness and pathogenesis. *Microbes and infection*, 105235, doi:10.1016/j.micinf.2023.105235 (2023).

Reviewer #2 (Remarks to the Author):

Antimicrobial resistance (AMR) is a global threat to public health.

The authors describe a new synthetic antibiotic class that is potentially effective against a number of clinically relevant bacteria.

The Authors' Response: We sincerely thank the Reviewer for the positive evaluation of our manuscript. The constructive comments and suggestions have helped us improve the quality

of our manuscript. We provide the point-by-point responses below.

1. **The Reviewer's Comment:** Results Section could be more effective if shortened to the most salient 5 to 6 Figs, focused on the design/synthesis; broad-spectrum antibacterial activity; minimal bacterial resistance; *in vivo* data; toxicity; mechanism of action.

The Authors' Response: We are extremely grateful for your constructive comments and advice. We deeply agree and appreciate your suggestion to shorten the 'Results' section and focus on the most salient 5 to 6 figures. We recognize that an excess of detail may obscure the main points and focus of the paper. In accordance with the Reviewer's suggestions, we have presented our findings in a more concise and focused manner, allowing readers to quickly grasp the core aspects of the study.

In the revised manuscript, we have adjusted and reorganized the six main figures to ensure they highlight key research findings effectively. Specifically included: **Fig 1. Discovery of F8. Fig 2. Antimicrobial activity of F8 *in vivo*. Fig 3. Safety evaluation of F8. Fig 4. F8 overcomes resistance and is effective *in vivo*. Fig 5. Binding of F8 with arcB. Fig 6. Bactericidal mechanism of F8.** Furthermore, we have meticulously revised the narration throughout the manuscript to align with the figures in each section. Specific modifications are marked in blue in the revised manuscript. For sections that have been condensed, we present them in detail in the Supplementary Material, allowing interested readers to explore our study further.

Please kindly review the changes in the revised manuscript. The quality of our manuscript has greatly improved thanks to your insights. We are profoundly grateful.

2. **The Reviewer's Comment:** Results jargon needs to be defined for a general audience.

The Authors' Response: We are extremely grateful for your constructive comments and advice. We sincerely apologize for our oversight on this matter. We recognize the importance of defining results jargon for a general audience, so we do our best to ensure that the terminology is understandable and universal. In accordance with the Reviewer's advice, we have carefully examined and supplemented these parts in the revised manuscript.

In the revised manuscript, we have amended the sentence from "We then modelled the binding pocket of the PTC region of the 50S subunit (method) and established an atomic property field" to "**We then modelled the binding pocket of the PTC region of the 50S subunit (method) and established an atomic property field that reflects preferences for various atomic properties at each point in space**".

In the revised manuscript, we inserted the content stating, "**Crash value is the degree of improper docking between the ligand and the receptor protein. Crash scores close to 0 are favorable**".

In the revised manuscript, we inserted the following content in **Figure legends**, "**Spleen index (mg/g) = spleen weight (mg)/body weight of mice (g)**" and "**Thymus index (mg/g) = thymus weight (mg)/body weight of mice (g)**".

Please kindly review the changes in the revised manuscript. The quality of our manuscript has greatly improved thanks to your insights. We are profoundly grateful.

3. **The Reviewer's Comment:** Discussion Section: repetitive, needs to focus on results in

context of prior work.

The Authors' Response: We are extremely grateful for your constructive comments and advice. We agree with the Reviewer's comments and have made adjustments and revisions to this section accordingly.

We have reviewed the entire manuscript, particularly the discussion section, removing all redundant analyses and simplifying the expression to ensure the discussion's conciseness and focus. At the same time, we have highlighted the results based on previous work, clearly pointing out and discussing each of the main findings. Furthermore, we have more explicitly emphasized the specific contributions of our study to the knowledge in the field. (Specific modifications are marked in blue in the revised manuscript.)

Please kindly review the changes in the **Discussion** of the revised manuscript. The quality of our manuscript has greatly improved thanks to your insights. We are profoundly grateful.

4. **The Reviewer's Comment:** line 28, Abstract: ..*in vitro* and *in vivo* broad-spectrum antibacterial activity against..

The Authors' Response: We are extremely grateful for your valuable suggestions. We sincerely apologize for our oversight on this matter. In accordance with the Reviewer's advice, we have amended the sentence from "The optimal modified compound, F8, was identified, which demonstrated excellent *in vitro* and *in vivo* anti-resistant bacterial activity and effectively mitigated the development of resistance" to "**The optimal modified compound, F8, was identified, which demonstrated excellent *in vitro* and *in vivo* broad-spectrum**

antibacterial activity against drug-resistant bacteria and effectively mitigated the development of resistance." Please kindly review the changes in the revised manuscript. The quality of our manuscript has greatly improved thanks to your insights. We are profoundly grateful.

5. **The Reviewer's Comment:** line 31, Abstract: In a mouse model of drug-resistant bacteremia...

The Authors' Response: We are extremely grateful for your valuable suggestions. In accordance with the Reviewer's advice, we have amended the sentence from "In the mouse model of drug-resistant bacterial bacteremia" to "**In a mouse model of drug-resistant bacteremia**". Please kindly review the changes in the revised manuscript. The quality of our manuscript has greatly improved thanks to your insights. We are profoundly grateful.

6. **The Reviewer's Comment:** lines 73-85: Move arcB para to Discussion.

The Authors' Response: We are extremely grateful for your constructive comments and advice. We sincerely apologize for our oversight on this matter. We have carefully reviewed the content in lines 73-85 and agree with the Reviewer's suggestion to move this section to the Discussion section. The content of lines 73-85 is repetitive with that of the Discussion section and contains in-depth explanations of the experimental results that might indeed be more appropriate in the Discussion section.

In accordance with the Reviewer's advice, we have moved lines 73-85 to the Discussion section. In the Discussion, we have integrated these contents, removing all redundancies to

ensure the discussion's conciseness and focus. (Specific modifications are marked in blue in the revised manuscript.) Please kindly review the changes in the **Discussion** of the revised manuscript. The quality of our manuscript has greatly improved thanks to your insights. We are profoundly grateful.

7. The Reviewer's Comment: line 74: ...biosynthesis and catabolism.

The Authors' Response: We are extremely grateful for your valuable suggestions. We sincerely apologize for our oversight on this matter. In accordance with the Reviewer's advice, we have amended the sentence from "biosynthesis and catabolism metabolism" to "**biosynthesis and catabolism**". Please kindly review the changes in the revised manuscript. The quality of our manuscript has greatly improved thanks to your insights. We are profoundly grateful.

8. The Reviewer's Comment: line 110: ...and *in vivo* antibacterial activity...

The Authors' Response: We are extremely grateful for your valuable suggestions. We sincerely apologize for our oversight on this matter. In accordance with the Reviewer's advice, we have amended the sentence from "We found that F8 exhibited excellent *in vitro* and *in vivo* anti-resistant bacterial activity" to "**We found that F8 exhibited excellent *in vitro* and *in vivo* antibacterial activity**". Please kindly review the changes in the revised manuscript. The quality of our manuscript has greatly improved thanks to your insights. We are profoundly grateful.

9. **The Reviewer's Comment:** line 114: Gram-positive and Gram-negative throughout.

The Authors' Response: We are extremely grateful for your constructive comments and advice. We are fully aware that our narration is imprecise and erroneous. In accordance with the Reviewer's advice, we carefully reviewed the entire manuscript and revised the sentence from "gram-positive and gram-negative bacteria" to "**Gram-positive and Gram-negative bacteria**". Please kindly review the changes in the revised manuscript. The quality of our manuscript has greatly improved thanks to your insights. We are profoundly grateful.

10. **The Reviewer's Comment:** line 142: define atomic property field.

The Authors' Response: We greatly appreciate your comprehensive and helpful critique. We have addressed the Reviewer's inquiries and have made revisions and additions to this section accordingly. The atomic property field reflects the preferences of various atomic properties at each point in space¹. Specifically, it involves attribute mapping or characteristic analysis at the atomic level within the binding pocket. This may be utilized for quantifying and describing the intermolecular forces between atoms or molecules, such as predicting the binding affinity between ligand molecules and protein targets in molecular docking studies^{2,3}. We have amended the sentence from "We then modelled the binding pocket of the PTC region of the 50S subunit (method) and established an atomic property field" to "**We then modelled the binding pocket of the PTC region of the 50S subunit (method) and established an atomic property field that reflects preferences for various atomic properties at each point in space**". Please kindly review the changes in the revised manuscript. The quality of our manuscript has greatly improved thanks to your insights. We are profoundly grateful.

11. **The Reviewer's Comment:** line 153: define crash value.

The Authors' Response: We greatly appreciate your comprehensive and helpful critique. We have addressed the Reviewer's inquiries and have made revisions and additions to this section accordingly. Crash value is the degree of inappropriate penetration by the ligand into the protein and of interpenetration between ligand atoms (self-clash) that are separated by rotatable bonds^{4,5}. In the revised manuscript, we inserted the content stating, "**Crash value is the degree of improper docking between the ligand and the receptor protein. Crash scores close to 0 are favorable**". Please kindly review the changes in the revised manuscript. The quality of our manuscript has greatly improved thanks to your insights. We are profoundly grateful.

12. **The Reviewer's Comment:** Tables 1-3: add legends.

The Authors' Response: We are extremely grateful for your constructive comments and advice. We sincerely apologize for our oversight on this matter. In accordance with the Reviewer's advice, we have inserted the following content in **Tables**. In **Table 1**, "**The MICs of antibiotics were determined by the broth microdilution method (see materials and methods) following the Clinical and Laboratory Standards Institute (CLSI) guidelines. The assay was done three times to confirm results (n = 3). Source data are provided as a Source Data file. MIC, Minimum Inhibitory Concentration; FLO, florfenicol**".

In **Table 2**, "**The MBCs of antibiotics were determined as the lowest concentration of tested compound that killed at least 99.9% of the initial inoculums. The assay was**

done three times to confirm results (n = 3). MBC, Minimum Bactericidal Concentration; FLO, florfenicol".

In Table 3, "The MICs of antibiotics were determined by the broth microdilution method (see materials and methods) following the Clinical and Laboratory Standards Institute (CLSI) guidelines. The assay was done three times to confirm results (n = 3). Source data are provided as a Source Data file. MIC, Minimum Inhibitory Concentration; FLO, florfenicol".

Please kindly review the changes in the revised manuscript. The quality of our manuscript has greatly improved thanks to your insights. We are profoundly grateful.

13. **The Reviewer's Comment:** line 193: F8 MIC of 128 uM for *Pseudomonas* is 2-fold below that shown for others.

The Authors' Response: We are extremely grateful for your valuable suggestions. In accordance with the Reviewer's advice, we have carefully examined this part in the manuscript. Upon reviewing our data, we confirm that the MIC of F8 for *P. aeruginosa* is 128 μ M.

This difference is attributed to the fact that *P. aeruginosa* is an opportunistic pathogen characterized by an innate resistance to multiple antimicrobials^{6,7}, a resistance that is increasingly attributable to the operation of broadly specific, tripartite multidrug efflux systems of the resistance-nodulation-division (RND) family⁸. In addition, florfenicol by itself is not an effective drug against *P. aeruginosa*, most strains of *P. aeruginosa* were intrinsically resistant to florfenicol due to very low permeability of their outer membrane and the efforts of

efflux pumps^{9,10}.

F8 is a compound with chemical structure modification of florfenicol, which has good antimicrobial activity against some bacteria, with a MIC of 128 μ M against the particularly challenging *P. aeruginosa*, which is better than florfenicol. Our research findings suggest that F8 has gained a second mode of action by targeting *arcB*, an ornithine carbamoyl transferase.

Although the effectiveness of F8 is improved compared to florfenicol, it is less effective against *P. aeruginosa* for the reasons mentioned above, including the fact that *P. aeruginosa* itself possesses a complex resistance mechanism and the fact that F8 may not have a significant effect on the main resistance mechanism of *P. aeruginosa*.

The quality of our manuscript has been greatly improved, thanks to your insights. We are immensely thankful for your guidance and support.

14. The Reviewer's Comment: Label figures 1-10.

The Authors' Response: We are extremely grateful for your constructive comments and advice. Regarding the labeling of the figures, we wish to clarify that in the original manuscript, detailed legends for all the figures (Figures 1 to 10) were indeed uniformly provided in a specific section of the document, located before all the figures and after the references. This structural arrangement was adopted because, according to the submission requirements, figures and legends need to be uploaded to the system separately: the legends accompany the original manuscript text, while the figures are uploaded independently to ensure their clarity.

In addition, based on your previous valuable suggestions, we have optimized and

reorganized the figures in the manuscript. The revised manuscript contains 6 carefully selected and reorganized figures instead of the original 10. In order to maintain the neatness and flow of the manuscript and ensure that readers can easily find detailed explanations and background information for each figure. Detailed legends for each figure have been uniformly placed in a specific section of the document, before all the figures and after the references for easy reference by the reader.

Please kindly review the changes in the revised manuscript. The quality of our manuscript has greatly improved thanks to your insights. We are profoundly grateful.

15. The Reviewer's Comment: Fig. 2 can be condensed to show principal *in vivo* data: CFU, and tissue data (Tables 1-3 *in vitro* data).

The Authors' Response: We are extremely grateful for your valuable suggestions, which are essential for us to present our research results accurately. In accordance with the Reviewer's advice, we have reorganized and adjusted the data in **Figure 2**, focusing solely on the key *in vivo* data, specifically colony-forming units (CFU) and tissue analysis. This adjustment enhances the clarity and impact of Figure 2, making it more intuitive and emphasizing the significant *in vivo* findings of our study. Additionally, we have revised the corresponding text sections to align with these changes in Figure 2. Please kindly review the changes in the revised manuscript. The quality of our manuscript has greatly improved thanks to your insights. We are profoundly grateful.

Fig 2. Antimicrobial activity of F8 *in vivo*. a–d, Bacterial load in the blood, liver, spleen, and kidney in the different treatment groups in a mouse *S. aureus* bacteraemia model (n = 6). ns, not significant, *, $p < 0.05$, **, $p < 0.01$, ***, $p < 0.001$. e, Histopathological assessment of the liver, spleen, and kidney in the different treatment groups in a mouse *S. aureus* bacteraemia model. Scale bar, 100 μm .

16. **The Reviewer's Comment:** Fig. 1c is a table that should be moved to supplementary material.

The Authors' Response: We are extremely grateful for your constructive comments and advice. We strongly agree with your suggestion that moving the table from Figure 1c to the Supplementary Material will make the body of the article more concise and enable readers to

focus more easily on the main findings and analysis. Acting on your recommendation, we have relocated the table originally in Figure 1c to the **Supplementary Table 1**. We believe that this adjustment not only improves the readability of the article but also ensures that all readers have easy access to the comprehensive data supporting our conclusions. Please kindly review the changes in the revised manuscript. The quality of our manuscript has greatly improved thanks to your insights. We are profoundly grateful.

Table S1. SYBYL simulation docking.

Name	Mol Wt	Total Score	Crash	Polar
F14	559	7.6589	-2.6455	5.0034
F8	497	6.1738	-1.9481	4.0442
F3	425	6.0496	-2.1376	3.6280
F1	453	5.9889	-1.1184	4.1346
F15	603	5.7043	-3.3284	2.1640
F4	440	5.6470	-0.9994	4.2054
F11	513	5.6375	-0.9141	2.3341
F2	514	5.4920	-1.0737	4.4195
F13	485	5.3018	-1.6204	3.4649
F7	475	5.2215	-2.1318	4.3768
FLO	358	5.1467	-2.2698	4.9464
F6	468	5.0195	-1.1588	3.3045

a. SYBYL simulation docking between 12 molecular structures and the PTC region of the 50s subunit. b. FLO, florfenicol.

17. The Reviewer's Comment: Fig. 3bc: define spleen index and thymus index.

The Authors' Response: We are extremely grateful for your valuable suggestions. We have modified the manuscript exactly according to your comments. In the revised manuscript, we inserted the following content in **Figure legends**, "**Spleen index (mg/g) = spleen weight (mg)/body weight of mice (g)**" and "**Thymus index (mg/g) = thymus weight (mg)/body weight of mice (g)**". Please kindly review the changes in the revised manuscript. The quality of our manuscript has greatly improved thanks to your insights. We are profoundly grateful.

18. **The Reviewer's Comment:** Fig. 3gh: for solid tissues, pg/mL should be expressed as pg/g.

The Authors' Response: We are extremely grateful for your constructive comments and advice. We have addressed the Reviewer's inquiries and have made revisions and additions to this section accordingly. In Fig. 3gh, we utilized the unit "pg/mL" to describe the concentration of specific solid tissue samples. This choice of unit is based on our experimental design, where 1 gram of solid tissue sample was indeed added to every 1 milliliter of solution. Therefore, in this specific scenario, "pg/mL" and "pg/g" are numerically equivalent. Recognizing the potential for confusion, we have corrected this oversight and have now expressed all pertinent data in "pg/g" to accurately convey the concentration measurements of the solid tissue. Please kindly review the changes in the revised manuscript. The quality of our manuscript has greatly improved thanks to your insights. We are profoundly grateful.

Fig 3. Safety evaluation of F8. **a**, Spleen index. Spleen index (mg/g) = spleen weight (mg)/body weight of mice (g). **b**, Thymus index. Thymus index (mg/g) = thymus weight (mg)/body weight of mice (g). **c**, Concentration of IL-6 in serum. **d**, Concentration of IL-2 in serum. **e**, Concentration of Hsp70 in serum. **f–h**, Concentration of Hsp70 in the spleen, thymus, and bone marrow. ns, not significant, *, $p < 0.05$, **, $p < 0.01$, ***, $p < 0.001$. **i**, Histopathological assessment of the spleen, thymus, and bone marrow. Scale bar, 100 μm . **j**, Apoptotic cells in the spleen, thymus, and bone marrow in different groups (TUNEL staining, under a fluorescence microscope at 200 \times). **k**, Acute toxicity test, record of mouse body weight after a single dose of F8 or positive control FLO at 5000 mg/kg (n=6). **l–n**, Toxicity of different concentrations of F8 or positive control FLO (0 – 256 μM) on the Vero cells, L0-2 cells and pk15 cells (n=6). **o**, Hemolysis rate of sheep red blood cells by different concentrations of F8 or positive control FLO (0 – 256 μM) (n=6).

19. **The Reviewer's Comment:** Fig. 4i: state route used for drug delivery.

The Authors' Response: We are extremely grateful for your valuable suggestions. We have modified the manuscript exactly according to your comments. In the revised manuscript, we inserted the following content in **Figure legends**, "**i, Survival curves of mice in an anti-*S. aureus* study (n = 10). F8/FLO were administered via oral gavage as the route of delivery.**" Please kindly review the changes in the revised manuscript. The quality of our manuscript has greatly improved thanks to your insights. We are profoundly grateful.

20. **The Reviewer's Comment:** Fig. 5-8 can be condensed/summarized or placed in Suppl.

The Authors' Response: We are extremely grateful for your valuable suggestions. We strongly agree with your suggestion that Figures 5-8 could be condensed and partially placed in the Supplementary Materials section, which would be very helpful in improving the reading experience of the paper and strengthening the presentation of the findings. In accordance with the Reviewer's advice, we carefully evaluated the data in Figures 5-8 to ensure that only the key data that most directly supported the conclusions of our study were

retained in the main figures. Figure 5a presents key analyses, which we adjusted and retained in order to highlight the main findings. Some of the non-primary outcomes or more indirect data were moved to the Supplementary Materials section. Additionally, we have revised the corresponding text sections to align with these changes in Figure 5-8. Please kindly review the changes in the revised manuscript. The quality of our manuscript has greatly improved thanks to your insights. We are profoundly grateful.

Fig 5. Binding of F8 with arcB. **a**, Experimental schematic for the multi-omics experiment. **b**, Gene-level differences of arcB in *S. aureus* treated with F8 or positive control FLO. Groups 1-3 represent three sets of transcriptomic samples, respectively. **c**, Simulated image of the F8 4Å DBD in arcB (PDB:2ksd) generated by PyMOL software, with residues LEU-31 shown in green. **d**, The superposition of the crystal structure of the DBDs of F8 and FLO bound to arcB, with residues LEU-31 shown in green, FLO in light purple, and F8 in blue. **e**, The expression

of arcB was analysed by SDS-PAGE, and compared with the negative control (C). The target protein was expressed in the form of total protein (T), soluble (S), and insoluble (P) of the lysate. **f, g**, ITC detection of ligand target-binding affinity. **h**, Detection of thermal displacement (T_m change) by DSF. **i**, MIC of F8 against strains with constitutive cfr resistance. **j**, MIC of F8 against the arcB deletion mutant. The assay was done three times to confirm results ($n = 3$). FLO, florfenicol.

21. The Reviewer's Comment: line 237: IL-2 and Hsp70 (Fig. 3ef) both show significant difference between PBS and F8 samples.

The Authors' Response: We are extremely grateful for your valuable suggestions. We sincerely apologize for our oversight on this matter and have accordingly modified the sentence. Specifically, ELISA results showed significant differences in blood levels of IL-6, IL-2, and Hsp70 between FLO-treated and control groups ($p < 0.001$). There was no significant difference in the blood levels of IL-6 between the F8-treated group and the control group, but there was a significant difference in the levels of IL-2 and Hsp70 ($p < 0.05$). There was a significant difference in blood levels of IL-6, IL-2 and Hsp70 between the F8-treated group and the FLO-treated group ($p < 0.01$).

In the revised manuscript, "The ELISA results showed no significant differences in the levels of IL-6, IL-2, and Hsp70 in the blood between the F8 treatment group and the control group." has been changed to "**ELISA results showed that there was no significant difference in the blood levels of IL-6 between the F8-treated group and the control group, but there was a significant difference in the levels of IL-2 and Hsp70 ($p < 0.05$).**" We deeply regret any mistakes in our writing and have taken steps to correct them. Please kindly review the changes in the revised manuscript. The quality of our manuscript has greatly improved thanks to your insights. We are profoundly grateful.

22. **The Reviewer's Comment:** line 245: HGB (Fig3o) shows significant difference between PBS and F8 samples.

The Authors' Response: We are extremely grateful for your valuable suggestions. We sincerely apologize for our oversight on this matter and have accordingly modified the sentence. In the revised manuscript, we have amended the sentence from "In addition, no significant differences in the number of white blood cells (WBCs), neutrophils (Neu), lymphocytes (Lym), red blood cells (RBCs), haemoglobin (HGB), and platelets (PLT) were observed between the F8-treated group and the control group, indicating that F8 has low toxicity to mice." to "**In addition, there was a significant difference in hemoglobin (HGB) between the F8-treated group and the control group. However, there were no significant differences in the number of white blood cells (WBCs), neutrophils (Neu), lymphocytes (Lym), red blood cells (RBCs), and platelets (PLT), indicating that F8 has low toxicity to mice.**" We deeply regret any mistakes in our writing and have taken steps to correct them. Please kindly review the changes in the revised manuscript. The quality of our manuscript has greatly improved thanks to your insights. We are profoundly grateful.

23. **The Reviewer's Comment:** line 249-253: toxicity data in supplemental Fig3a, Fig4a-c, Fig5a belongs in main figs.

The Authors' Response: We are extremely grateful for your constructive comments and advice. We fully agree with including these key toxicity data in the main figures, which is crucial for readers to understand our research findings. In accordance with the Reviewer's

advice, we have integrated the toxicity data from Supplementary Figures 3a, 4a-c, and 5a into **Figure 3**. This adjustment aims to present important data more visually, ensuring the integrity of the results while enhancing the readability of the data. Additionally, we have revised the corresponding text sections to align with these changes in the Figures. Please kindly review the changes in the revised manuscript. The quality of our manuscript has greatly improved thanks to your insights. We are profoundly grateful.

Fig 3. Safety evaluation of F8. **a**, Spleen index. Spleen index (mg/g) = spleen weight (mg)/body weight of mice (g). **b**, Thymus index. Thymus index (mg/g) = thymus weight (mg)/body weight of mice (g). **c**, Concentration of IL-6 in serum. **d**, Concentration of IL-2 in serum. **e**, Concentration of Hsp70 in serum. **f-h**, Concentration of Hsp70 in the spleen, thymus, and bone marrow. ns, not significant, *, $p < 0.05$, **, $p < 0.01$, ***, $p < 0.001$. **i**, Histopathological assessment of the spleen, thymus, and bone marrow. Scale bar, 100 μm . **j**, Apoptotic cells in the spleen, thymus, and bone marrow in different groups (TUNEL staining, under a fluorescence microscope at 200 \times). **k**, Acute toxicity test, record of mouse body weight after a single dose of F8 or positive control FLO at 5000 mg/kg (n=6). **l-n**, Toxicity of different concentrations of F8 or positive control FLO (0 – 256 μM) on the Vero cells, L-02 cells and pk15 cells (n=6). **o**, Hemolysis rate of sheep red blood cells by different concentrations of F8 or positive control FLO (0 – 256 μM) (n=6).

24. **The Reviewer's Comment:** lines 319-321: repetitive from 317-319.

The Authors' Response: We are extremely grateful for your valuable suggestions and sincerely apologize for our oversight on this matter. In accordance with the Reviewer's advice, we have eliminated the text from lines 319-321 of the original manuscript: "We speculate that the antibacterial activity of F8 is associated with multiple pathways, including the arginine degradation metabolic pathway, the bacterial cell membrane, and energy metabolism," to address the issue of unnecessary repetition. Please kindly review the changes in the revised manuscript. The quality of our manuscript has greatly improved thanks to your insights. We are profoundly grateful.

25. **The Reviewer's Comment:** lines 327: Fig. 9c is referred to before (Fig. 9a,b).

The Authors' Response: We are extremely grateful for your valuable suggestions and sincerely apologize for our oversight on this matter. In accordance with the Reviewer's advice, we have rearranged the sequence of Figure 9 such that Figure 9c now precedes Figures 9a and 9b, to better reflect the logical order of the research content. Additionally, with the change in the sequence of the figures in the manuscript, we have correspondingly adjusted the main text to ensure the fluency and coherence of the paper. Please kindly review the changes in the revised manuscript. The quality of our manuscript has greatly improved thanks to your insights. We are profoundly grateful.

26. **The Reviewer's Comment:** label Fig. 9c w/gene expression; define groups 1-3 in legend.

The Authors' Response: We are extremely grateful for your constructive comments and

advice. We have modified the manuscript exactly according to your comments. In accordance with the Reviewer's advice, we revised the labels for Fig. 9c. "Relative fold change" has been changed to "**w/gene expression**".

Furthermore, we have defined groups 1-3 in the legend accompanying Fig. 9c to ensure that the experimental design and group categorizations are transparent and easily understandable for all readers. In the revised manuscript, we inserted the following content in Figure legends, "**Groups 1-3 represent three sets of transcriptomic samples, respectively.**" Please kindly review the changes in the revised manuscript. The quality of our manuscript has greatly improved thanks to your insights. We are profoundly grateful.

Fig 5. Binding of F8 with arcB. b, Gene-level differences of arcB in *S. aureus* treated with F8 or positive control FLO. Groups 1-3 represent three sets of transcriptomic samples, respectively.

27. **The Reviewer's Comment:** Fig. 9d: state composition of the negative control; indicate expected size of ArcB.

The Authors' Response: We are extremely grateful for your valuable suggestions and sincerely apologize for our oversight on this matter. The negative control (C) was an empty pET28a vector, which does not contain any inserted target gene fragment. The expected size

of arcB is 37.74 kDa. In the revised manuscript, we inserted the following content in Figure legends, "**The expression of arcB was analysed by SDS-PAGE, and compared with the negative control (C, empty pET28a vector). The target protein was expressed in the form of total protein (T), soluble (S), and insoluble (P) of the lysate. The expected size of arcB is 37.74 kDa.**" Please kindly review the changes in the revised manuscript. The quality of our manuscript has greatly improved thanks to your insights. We are profoundly grateful.

Fig 5. Binding of F8 with arcB. e, The expression of arcB was analysed by SDS-PAGE, and compared with the negative control (C, empty pET28a vector). The target protein was expressed in the form of total protein (T), soluble (S), and insoluble (P) of the lysate. The expected size of arcB is 37.74 kDa.

28. The Reviewer's Comment: Fig. 9g typo: blank-arcB.

The Authors' Response: We are extremely grateful for your valuable suggestions. We sincerely apologize for our oversight on this matter and have accordingly modified the sentence. "black-arcB" has been changed to "**blank-arcB**". We deeply regret any mistakes in our writing and have taken steps to correct them. Please kindly review the changes in the revised manuscript. The quality of our manuscript has greatly improved thanks to your insights. We are profoundly grateful.

Fig 5. Binding of F8 with arcB. h, Detection of thermal displacement (T_m change) by DSF.

29. **The Reviewer's Comment:** lines 364-366: define permeability by PI staining; Fig. 10c,d; label with PI staining.

The Authors' Response: We are extremely grateful for your constructive comments and advice. We have modified the manuscript exactly according to your comments. In the revised manuscript, we have amended the sentence from "Further examination of bacterial cytoplasmic membrane permeability showed that F8 induced increased membrane permeability in *S. aureus*" to "**Further examination of bacterial cytoplasmic membrane permeability by PI staining showed that F8 induced increased membrane permeability in *S. aureus***".

In accordance with the Reviewer's advice, we revised the labels for Fig. 10c, d. "Fluorescence intensity" has been changed to "**PI staining**". Please kindly review the changes in the revised manuscript. The quality of our manuscript has greatly improved thanks to your insights. We are profoundly grateful.

Fig 6. Bactericidal mechanism of F8. **a, b,** Concentrations of extracellular DNA, RNA, and protein at different time intervals after treatment with various concentrations (4×MIC, 2×MIC, and MIC) of F8 in *S. aureus* and *E. coli*. **c, d,** Increased membrane permeability in *S. aureus* and *E. coli* after treatment with 4×MIC of F8. Membrane permeability measured by propidium iodide (PI), with excitation/emission wavelengths at 535 nm/615 nm. **e, f,** Dissipation of ΔpH by F8 and FLO in *S. aureus* and *E. coli*. Exponential-phase *S. aureus* and *E. coli* were incubated with pH fluorescence probe BCECF-AM. The cells were treated with 4×MIC of F8 and FLO, and the intracellular pH was determined. Fluorescence measurements were taken at 5-minute intervals using an excitation/emission wavelength of 488 nm/535 nm. **g, h,** Accumulation of intracellular ROS in *S. aureus* and *E. coli* after treatment with 4×MIC of F8 and FLO. The fluorescence probe DCFH-DA (10 μmol/L) was added to the bacterial suspension. The cells were treated with 4×MIC of F8 and FLO. After incubation for 2 h, the fluorescent values were measured with an excitation/emission wavelength of 488 nm/525 nm. ROSUP was used as the positive control. **i, j,** Changes in intracellular ATP levels in *S. aureus* and *E. coli* after treatment with F8 and FLO. After treatment with 4×MIC of FLO or F8 for 2 h, bacterial cells were lysed using lysozyme, and following centrifugation, the supernatant was collected to measure the intracellular ATP levels. Luminescence was measured using the Infinite M200 Microplate reader (Tecan). ns, not significant, *, $p < 0.05$, **, $p < 0.01$, ***, $p < 0.001$.

30. **The Reviewer's Comment:** Fig. 10ef: Explain pH measurement; label w/pH.

The Authors' Response: We are extremely grateful for your constructive comments and advice. We have modified the manuscript exactly according to your comments. In the revised manuscript, we inserted the following content in **Figure legends**, "**e, f, Dissipation of Δ pH by F8 and FLO in *S. aureus* and *E. coli*. Exponential-phase *S. aureus* and *E. coli* were incubated with pH fluorescence probe BCECF-AM. The cells were treated with 4×MIC of F8 and FLO, and the intracellular pH was determined. Fluorescence measurements were taken at 5-minute intervals using an excitation/emission wavelength of 488 nm/535 nm.**"

In accordance with the Reviewer's advice, we revised the labels for Fig. 10e, f. "Fluorescence intensity" has been changed to "**w/pH**". Please kindly review the changes in the revised manuscript. The quality of our manuscript has greatly improved thanks to your insights. We are profoundly grateful.

31. **The Reviewer's Comment:** Fig. 10gh: Explain ROS measurement; label w/ROS.

The Authors' Response: We are extremely grateful for your constructive comments and advice. We have modified the manuscript exactly according to your comments. In the revised manuscript, we inserted the following content in **Figure legends**, "**g, h, Accumulation of intracellular ROS in *S. aureus* and *E. coli* after treatment with 4×MIC of F8 and FLO. The fluorescence probe DCFH-DA (10 μ mol/L) was added to the bacterial suspension. The cells were treated with 4×MIC of F8 and FLO. After incubation for 2 h, the**

fluorescent values were measured with an excitation/emission wavelength of 488 nm/525 nm. ROSUP was used as the positive control."

In accordance with the Reviewer's advice, we revised the labels for Fig. 10e, f. "Fluorescence intensity" has been changed to "**w/ROS**". Please kindly review the changes in the revised manuscript. The quality of our manuscript has greatly improved thanks to your insights. We are profoundly grateful.

32. The Reviewer's Comment: Fig. 10ij: Explain ATP measurement.

The Authors' Response: We are extremely grateful for your constructive comments and advice. We have modified the manuscript exactly according to your comments. In the revised manuscript, we inserted the following content in **Figure legends, "i, j, Changes in intracellular ATP levels in *S. aureus* and *E. coli* after treatment with F8 and FLO. After treatment with 4×MIC of FLO or F8 for 2 h, bacterial cells were lysed using lysozyme, and following centrifugation, the supernatant was collected to measure the intracellular ATP levels. Luminescence was measured using the Infinite M200 Microplate reader (Tecan).**" Please kindly review the changes in the revised manuscript. The quality of our manuscript has greatly improved thanks to your insights. We are profoundly grateful.

References

- 1 Totrov, M. Atomic property fields: generalized 3D pharmacophoric potential for automated ligand superposition, pharmacophore elucidation and 3D QSAR. *Chemical biology & drug design* **71**, 15-27, doi:10.1111/j.1747-0285.2007.00605.x (2008).
- 2 Totrov, M. Ligand binding site superposition and comparison based on Atomic

- Property Fields: identification of distant homologues, convergent evolution and PDB-wide clustering of binding sites. *BMC bioinformatics* **12 Suppl 1**, S35, doi:10.1186/1471-2105-12-s1-s35 (2011).
- 3 Ehrt, C., Brinkjost, T. & Koch, O. Impact of Binding Site Comparisons on Medicinal Chemistry and Rational Molecular Design. *Journal of medicinal chemistry* **59**, 4121-4151, doi:10.1021/acs.jmedchem.6b00078 (2016).
- 4 Vyas, V. K., Goel, A., Ghate, M. & Patel, P. Ligand and structure-based approaches for the identification of SIRT1 activators. *Chemico-biological interactions* **228**, 9-17, doi:10.1016/j.cbi.2015.01.001 (2015).
- 5 Rostom, S. A. F., Badr, M. H., Abd El Razik, H. A. & Ashour, H. M. A. Structure-based development of novel triazoles and related thiazolotriazoles as anticancer agents and Cdc25A/B phosphatase inhibitors. Synthesis, in vitro biological evaluation, molecular docking and in silico ADME-T studies. *European journal of medicinal chemistry* **139**, 263-279, doi:10.1016/j.ejmech.2017.07.053 (2017).
- 6 Hancock, R. E. & Speert, D. P. Antibiotic resistance in *Pseudomonas aeruginosa*: mechanisms and impact on treatment. *Drug resistance updates : reviews and commentaries in antimicrobial and anticancer chemotherapy* **3**, 247-255, doi:10.1054/drup.2000.0152 (2000).
- 7 Breidenstein, E. B., de la Fuente-Núñez, C. & Hancock, R. E. *Pseudomonas aeruginosa*: all roads lead to resistance. *Trends in microbiology* **19**, 419-426, doi:10.1016/j.tim.2011.04.005 (2011).
- 8 Poole, K. Efflux-mediated antimicrobial resistance. *The Journal of antimicrobial chemotherapy* **56**, 20-51, doi:10.1093/jac/dki171 (2005).
- 9 Angus, B. L., Carey, A. M., Caron, D. A., Kropinski, A. M. & Hancock, R. E. Outer membrane permeability in *Pseudomonas aeruginosa*: comparison of a wild-type with an antibiotic-supersusceptible mutant. *Antimicrobial agents and chemotherapy* **21**, 299-309, doi:10.1128/aac.21.2.299 (1982).
- 10 Li, X. Z., Plésiat, P. & Nikaido, H. The challenge of efflux-mediated antibiotic resistance in Gram-negative bacteria. *Clinical microbiology reviews* **28**, 337-418, doi:10.1128/cmr.00117-14 (2015).

Reviewer #3 (Remarks to the Author):

Noteworthy Results: The manuscript presents the development of a new antibiotic, F8, effective against various antibiotic-resistant bacteria. The identification of a novel target, ornithine carbamoyl transferase (arcB), and the use of multi-omics analysis are key highlights.

Significance to the Field: The work is significant due to its potential impact on addressing antibiotic resistance, a major global health issue. The approach to drug design and target identification could influence future research in the field.

Comparison to Established Literature: The manuscript appears original in its approach to combining structural hybridization-based drug design with multi-omics analysis for target identification. This approach is distinct from traditional methods in antibiotic research.

Support for Conclusions: The conclusions are supported by comprehensive data, including in vitro and in vivo studies, molecular docking, and biochemical assays. However, further research and clinical trials might be needed to fully validate the drug's effectiveness and safety.

Flaws in Data Analysis or Interpretation: The analysis and interpretation of data appear robust. However, as with any scientific study, independent replication and additional studies could further validate the findings.

Soundness of Methodology: The methodology, encompassing drug design, target identification, and validation, are sound and adheres to current standards in the field.

Detail for Reproduction: The manuscript provides sufficient detail for the methods, allowing for potential reproduction of the research, which is crucial for scientific validation.

The Authors' Response: We sincerely thank the Reviewer for the positive evaluation of our manuscript. The constructive comments and suggestions have helped us improve the quality of our manuscript. We provide the point-by-point responses below.

1. **The Reviewer's Comment:** Expanded Discussion on Limitations: More detailed discussion on potential limitations of F8, including any observed side effects, resistance development potential, and effectiveness in diverse biological environments.

The Authors' Response: We are extremely grateful for your constructive comments and advice. In response to the Reviewer's valuable suggestion for a more detailed exploration of F8's potential limitations, we have enriched the discussion in our revised manuscript. Specifically, we inserted the following content in **Discussion**, "**Given that F8 exhibits significant antimicrobial activity and has the potential to effectively mitigate the development of resistance, further extensive work is needed to expand its antibacterial spectrum against both standard and resistant strains, as well as to assess its efficacy in diverse biological environments. Although the structural modification strategy is crucial and effective, the resulting compounds are ultimately influenced by the same type of resistance mechanism, which evolves and spreads in response to the resistance mechanism of previously-used compounds. Our study suggests that F8 targets the arcB gene, which may result in novel mechanisms of antibacterial action. However, we do not rule out the possibility that F8 may also affect other biological pathways and molecular**

targets. For example, it is important to determine whether F8 is susceptible to resistance mechanisms associated with phenicols in clinical settings, as this could influence the development of resistance. Our study preliminarily investigated the safety profile of F8, encompassing acute toxicity, organ damage, immunotoxicity, and hematotoxicity. Future research will need to delve deeper and broaden the scope of investigation into its side effects, including long-term toxicity, genotoxicity, and other potential adverse effects. In addition, current environmental contamination from antibiotics is a matter of concern. It is particularly crucial to conduct an environmental risk assessment for F8, which should include its degradability in aquatic and soil environments, the assessment of health risks to the biotic community, and the potential transfer of resistance genes."

(Specific modifications are marked in blue in the revised manuscript.) Please kindly review the changes in the Discussion of the revised manuscript. The quality of our manuscript has greatly improved thanks to your insights. We are profoundly grateful.

2. The Reviewer's Comment: Comparison with Existing Antibiotics: Provide a more comprehensive comparison of F8 with existing antibiotics, particularly those targeting similar bacteria, to contextualize its efficacy and potential advantages.

The Authors' Response: We are extremely grateful for your valuable suggestions. The need for a more comprehensive comparison with existing antibiotics, as you have pointed out, is an important aspect of evaluating the efficacy of F8. We sincerely appreciate your suggestion and understand its significance for enhancing the depth and breadth of our research.

F8 is derived from structural modifications to florfenicol. We focused on the difference

in antimicrobial activity between F8 and florfenicol. We decided at the initial stage to concentrate our resources on the effect of F8 against drug-resistant bacteria in order to gain a quick understanding of its potential and the mechanisms involved. This focus was also intended to provide preliminary evidence of its efficacy, setting the groundwork for more comprehensive comparative studies in subsequent phases.

Preliminary data indicate that F8 exhibits promising prospects for drug-resistant bacteria. Therefore, we have directed special attention to the effects of F8 on strains resistant to some significant antibiotics. Our research findings indicate that F8 has a strong inhibitory effect on methicillin-resistant *S. aureus*, with a MIC value of 8 μ M, consistent with the results observed in the standard strains. Polymyxins, particularly polymyxin B, have become the last line of therapy against multi-drug resistant Gram-negative bacteria. However, the rapid development of resistance has led to the emergence of more and more polymyxin B resistant bacteria. The MIC value of F8 against polymyxin B-resistant *E. hormaechei* is 16 μ M, which is a promising lead compound for resisting polymyxin B resistance. Sulfamethoxazole is reported to be one of the most widely used sulfonamide antibiotics in the world; however, the increasingly drug resistance greatly limits its utility. The MIC value of F8 against sulfamethoxazole-resistant *S. typhi* only 4 μ M. In addition, F8 also showed excellent bactericidal effects on FLO-resistant *S. suis*, FLO-resistant *H. parasuis* and doxycycline-resistant *S. typhi*. We have carefully reviewed and supplemented the content related to this section throughout the entire text (Specific modifications are marked in blue in the revised manuscript).

We greatly appreciate your valuable suggestions. We believe that an in-depth discussion of these issues will not only strengthen the conclusions of the current study, but also guide our

future work. We look forward to receiving your further guidance and suggestions.

3. The Reviewer's Comment: Environmental Impact Assessment: Discuss any potential environmental impacts of F8, particularly if used widely, considering the ongoing concerns about antibiotic pollution.

The Authors' Response: We are extremely grateful for your constructive comments and advice. We sincerely apologize for our oversight on this matter. We are acutely aware of the importance of environmental protection, especially in the context of current global challenges such as antibiotic pollution. To better address your concerns, we have briefly supplemented the discussion section with the potential environmental impacts caused by F8. In the revised manuscript, we inserted the following content in **Discussion**, "**Our study preliminarily investigated the safety profile of F8, encompassing acute toxicity, organ damage, immunotoxicity, and hemotoxicity. Future research will need to delve deeper and broaden the scope of investigation into its side effects, including long-term toxicity, genotoxicity, and other potential adverse effects. In addition, current environmental contamination from antibiotics is a matter of concern. It is particularly crucial to conduct an environmental risk assessment for F8, which should include its degradability in aquatic and soil environments, the assessment of health risks to the biotic community, and the potential transfer of resistance genes.**" (Specific modifications are marked in blue in the revised manuscript.) Please kindly review the changes in the Discussion of the revised manuscript. The quality of our manuscript has greatly improved thanks to your insights. We are profoundly grateful.

4. **The Reviewer's Comment:** Mechanism of Resistance Development: Could there be a discussion on how bacteria might develop resistance to F8, as understanding resistance mechanisms is key to developing more effective antibiotics.

The Authors' Response: We are extremely grateful for your valuable suggestions. We have addressed the Reviewer's inquiries and have made revisions and additions to this section accordingly. In the revised manuscript, we inserted the following content in **Discussion**, **"Although the structural modification strategy is crucial and effective, the resulting compounds are ultimately influenced by the same type of resistance mechanism, which evolves and spreads in response to the resistance mechanism of previously-used compounds. Our study suggests that F8 targets the arcB gene, which may result in novel mechanisms of antibacterial action. However, we do not rule out the possibility that F8 may also affect other biological pathways and molecular targets. For example, it is important to determine whether F8 is susceptible to resistance mechanisms associated with phenicols in clinical settings, as this could influence the development of resistance."**

In addition, we have carefully adjusted and supplemented the content related to this section throughout the entire text (Specific modifications are marked in blue in the revised manuscript). Please kindly review the changes in the revised manuscript. The quality of our manuscript has greatly improved thanks to your insights. We are profoundly grateful.

REVIEWERS' COMMENTS

Reviewer #1 (Remarks to the Author):

The authors have thoughtfully and thoroughly addressed the majority of my concerns in this revised version of the manuscript. I suggest only a few minor changes, and after they are made, I think this will be an excellent manuscript for Nature Communications.

- "designed through drug structural modification-based structure-guided design" seems long and multiply redundant (structure and designed are mentioned twice!). I suggest: "... we report development of a modified antimicrobial drug through structure-based drug design (SBDD) and modular synthesis."
- Overall, the use of "structural modification" throughout the manuscript as a direct replacement for structure hybridization is distracting. Structural modification is one of the oldest ways to optimize drugs... discussing it as a unique or novel approach doesn't really make sense. Most antibiotics on the market were made with structural modification. Basically, smashing a new term into the old structure doesn't quite work... the discussion of combination therapy tied to hybrid antibiotics makes sense, but tied to structure modification, which is pretty much all of drug discovery, doesn't quite make sense. The quality of the science in the manuscript is so good that this flaw in the opening discussion serves as a distraction.
- I think the authors may mitigate this awkwardness partially by discussing how structure-based drug design on the ribosome has been challenging because of low resolution structures and lack of computational tools that effectively handle large RNA-based binding sites. Then naturally evolve the discussion to how modern structural techniques and new computational methods have enabled the current approach, and revitalized amphenicol research. In some cases, you can replace "structural modification strategy" with SBDD and modular synthesis. In many other cases, this term can be simply removed altogether to lead to more impactful sentences about the results.

Title: A synthetic antibiotic class with a deeply-optimized design for overcoming bacterial resistance

The following is a point-to-point response to the Reviewers' comments.

Reviewer #1 (Remarks to the Author):

The authors have thoughtfully and thoroughly addressed the majority of my concerns in this revised version of the manuscript. I suggest only a few minor changes, and after they are made, I think this will be an excellent manuscript for Nature Communications.

The Authors' Response: We sincerely thank the Reviewer for the positive evaluation of our manuscript. The constructive comments and suggestions have helped us improve the quality of our manuscript. We provide the point-by-point responses below.

1. **The Reviewer's Comment:** "designed through drug structural modification-based structure-guided design" seems long and multiply redundant (structure and designed are mentioned twice!). I suggest: "... we report development of a modified antimicrobial drug through structure-based drug design (SBDD) and modular synthesis."

The Authors' Response: We are extremely grateful for your valuable suggestions and sincerely apologize for our oversight on this matter. We highly value your feedback and have revised the original sentence to improve the readability and professionalism of the paper. In accordance with the Reviewer's advice, we have amended the sentence from "Based on this study, we report the development of a modified antimicrobial drug that is rationally designed through drug structural modification-based structure-guided design and component-based

synthesis" to "**Based on this study, we report development of a modified antimicrobial drug through structure-based drug design (SBDD) and modular synthesis.**" (Specific modifications are marked in blue in the revised manuscript.) This revision not only makes the sentence more concise but also better emphasizes the core methods and innovations of our research. Please kindly review the changes in the Abstract of the revised manuscript. The quality of our manuscript has greatly improved thanks to your insights. We are profoundly grateful for your expertise and thoughtful feedback.

2. The Reviewer's Comment: Overall, the use of "structural modification" throughout the manuscript as a direct replacement for structure hybridization is distracting. Structural modification is one of the oldest ways to optimize drugs... discussing it as a unique or novel approach doesn't really make sense. Most antibiotics on the market were made with structural modification. Basically, smashing a new term into the old structure doesn't quite work... the discussion of combination therapy tied to hybrid antibiotics makes sense, but tied to structure modification, which is pretty much all of drug discovery, doesn't quite make sense. The quality of the science in the manuscript is so good that this flaw in the opening discussion serves as a distraction.

The Authors' Response: We are extremely grateful for your constructive comments and advice. We agree with the Reviewer's comments and have made adjustments and revisions to this section accordingly.

As you rightly pointed out, the use of "structural modification" throughout the manuscript as a direct replacement for "structure hybridization" was indeed misleading. We acknowledge that structural modification is a well-established approach in drug optimization

and should not be presented as a novel strategy. To rectify this, we have revised the terminology in our manuscript to more accurately reflect the scientific approaches discussed. Specifically, we have replaced mentions of "structural modification" with "SBDD and modular synthesis" where appropriate and ensured that each term is used correctly in the context of our study.

We removed the description of structural modifications and discussed that structure-based drug design on ribosomes is challenging. In accordance with the Reviewer's advice, we inserted the content stating, **"Structure-based drug design (SBDD) on the ribosome has encountered significant challenges, including low-resolution structural data and the lack of computational tools capable of effectively handling large RNA-based binding sites^{15,16,17}. Early structural studies of the ribosome provided limited details, making it difficult to accurately model drug interactions and predict binding affinities. Before resolving the structures of ribosomal subunits and their complexes with known antibiotics, computational design of novel antibacterial agents focused primarily on ligands. These issues once hindered the progress of amphenicols' research. However, advancements in modern structural techniques and the development of new computational methods have now made structure-based drug design feasible, thereby greatly revitalizing this field of research. Furthermore, compared to combination therapy, SBDD better addresses issues of differential bioavailability, pharmacokinetics, and metabolism, resulting in improved therapeutic safety and avoidance of drug-drug interactions."** (Specific modifications are marked in blue in the revised manuscript.)

Please kindly review the changes in the Abstract of the revised manuscript. The quality

of our manuscript has greatly improved thanks to your insights. We are profoundly grateful for your expertise and thoughtful feedback.

3. The Reviewer's Comment: I think the authors may mitigate this awkwardness partially by discussing how structure-based drug design on the ribosome has been challenging because of low resolution structures and lack of computational tools that effectively handle large RNA-based binding sites. Then naturally evolve the discussion to how modern structural techniques and new computational methods have enabled the current approach, and revitalized amphenicol research. In some cases, you can replace "structural modification strategy" with SBDD and modular synthesis. In many other cases, this term can be simply removed altogether to lead to more impactful sentences about the results.

The Authors' Response: We are extremely grateful for your constructive comments and advice. Your insights are critically important for the deepening and refinement of our research. We agree with the Reviewer's comments and have made adjustments and revisions to this section accordingly.

In the revised manuscript, we inserted the content stating, "**Structure-based drug design (SBDD) on the ribosome has encountered significant challenges, including low-resolution structural data and the lack of computational tools capable of effectively handling large RNA-based binding sites^{15,16,17}. Early structural studies of the ribosome provided limited details, making it difficult to accurately model drug interactions and predict binding affinities. Before resolving the structures of ribosomal subunits and their complexes with known antibiotics, computational design of novel antibacterial agents focused primarily on ligands. These issues once hindered the progress of amphenicols' research. However, advancements in modern structural techniques and the development of new computational methods have now made structure-based drug design feasible,**

thereby greatly revitalizing this field of research. Furthermore, compared to combination therapy, SBDD better addresses issues of differential bioavailability, pharmacokinetics, and metabolism, resulting in improved therapeutic safety and avoidance of drug-drug interactions." (Specific modifications are marked in blue in the revised manuscript.)

In line with your suggestions, we have replaced the term "structural modification strategy" with "SBDD" and "modular synthesis" where appropriate. In several instances, we have also removed the term altogether to improve the clarity and impact of our results. These changes have been made to ensure the terminology is precise and fitting within the context of our study.

Please kindly review the changes in the Abstract of the revised manuscript. The quality of our manuscript has greatly improved thanks to your insights. We are profoundly grateful for your expertise and thoughtful feedback.